# Modelling genetic mosaicism of neurodevelopmental disorders in vivo by a Cre-amplifying fluorescent reporter

Francesco Trovato[1,10] ✉, Riccardo Parra [1,10], Enrico Pracucci [1], Silvia Landi[1,2], Olga Cozzolino [1], Gabriele Nardi[1], Federica Cruciani[1], Vinoshene Pillai[1], Laura Mosti [1], Andrzej W. Cwetsch[3,4], Laura Cancedda [3,5], Laura Gritti[6], Carlo Sala[6], Chiara Verpelli[6], Andrea Maset[7,8], Claudia Lodovichi[7,8,9] & Gian Michele Ratto [1] ✉

Genetic mosaicism, a condition in which an organ includes cells with different genotypes, is frequently present in monogenic diseases of the central nervous system caused by the random inactivation of the X-chromosome, in the case of X-linked pathologies, or by somatic mutations affecting a subset of neurons. The comprehension of the mechanisms of these diseases and of the cell-autonomous effects of specific mutations requires the generation of sparse mosaic models, in which the genotype of each neuron is univocally identified by the expression of a fluorescent protein in vivo. Here, we show a dual-color reporter system that, when expressed in a floxed mouse line for a target gene, leads to the creation of mosaics with tunable degree. We demonstrate the generation of a knockout mosaic of the autism/epilepsy related gene PTEN in which the genotype of each neuron is reliably identified, and the neuronal phenotype is accurately characterized by two-photon microscopy.

[1] National Enterprise for Nanoscience and Nanotechnology (NEST), Istituto Nanoscienze Consiglio Nazionale delle Ricerche (CNR) and Scuola Normale Superiore Pisa, 56127 Pisa, Italy. [2] Institute of Neuroscience CNR, Pisa, Italy. [3] Istituto Italiano di Tecnologia, Genoa, Italy. [4] Università degli studi di Genova, Genoa, Italy. [5] Istituto Telethon Dulbecco, Rome, Italy. [6] Institute of Neuroscience CNR, Milan, Italy. [7] Veneto Institute of Molecular Medicine, Padua, Italy. [8] Padova Neuroscience Center, Padova Università di Padova, Padua, Italy. [9] Institute of Neuroscience CNR, Padua, Italy. [10] These authors contributed equally: Francesco Trovato, Riccardo Parra. ✉email: francesco.trovato@sns.it; gianmichele.ratto@sns.it

Genetic mosaicism refers to the presence of genetically distinct cellular populations within the same individual. This condition originates from DNA mutations, either monogenic or genome-wide, which may occur through different biological mechanisms, and has been associated with several pathologies[1–3].

Mosaicism of genes located on the X-chromosome naturally occurs in females because of the random inactivation of one copy of the X-chromosome. In this case, although all cells have the same genome, only one of the two alleles remains active in each cell, resulting in a whole-body scattered genetic mosaic[4]. Examples of dominant X-linked disorders are Fragile X syndrome, typically caused by a triplet expansion in the FMR1 gene on the X chromosome, Rett syndrome, a brain disorder caused by genetic mutations affecting MeCP2 protein function and lissencephaly, caused by mutations of the DCX gene[5,6]. One remarkable example of how mosaicism plays a key role in defining the endophenotype of the disease is PCDH19 girl cluster epilepsy, an emerging syndrome, associated with cognitive and sensory deficiencies caused by a mosaic of expression of the protocadherin 19 gene[7]. The disease affects only heterozygous females (showing a mosaic of wild type (WT) and mutated cells), while hemizygous males are not affected.

A pathophysiological role for somatic mosaicism has been initially identified for disorders associated to a disruption of cortical layout caused by erroneous migration[8,9] leading to focal cortical dysplasia[10] or, in extreme cases, to hemimegalencephaly, in which an entire hemisphere is dysplastic[11,12]. As our capacity of detecting somatic mosaicism in human patients improved, it gradually emerged that somatic mosaicism plays an important role in brain pathology, in which even low frequency of mutation can lead to cognitive disorders and epilepsy[3]. Genes involved in these processes include not only factors leading to alteration of cortical lamination, but also a wide spectrum of transcripts including ionic channels, adhesion molecules and elements of the synaptic structure[3]. In all cases, affected individuals may present a broad range of clinical manifestations, such as developmental delay, moderate to severe intellectual disability, autistic features and comorbid epilepsy. To underline the importance of this mechanism, recent whole-exome studies on genetic samples from families including one autism spectrum disorder (ASD) offspring have detected the presence of somatic mosaicism in 3–5% of the analyzed samples[13–16]. Somatic mutations can also result in mosaicism in males for X-linked genes. Interestingly, male patients with somatic mosaicism of PCDH19 present clinical manifestations identical to affected girls[7,17,18], strongly suggesting that mosaicism itself plays a pivotal role in determining this disease[7,19].

Given this background, it is not surprising that mosaic modeling has attracted wide interest, since it would allow the study of single-cell functions and cell-autonomous effects of selective overexpression/knockout (KO) in distinct cohorts of mutant and WT cells intermingled in the same environment. This approach is often more precise and informative than the use of classical genetics, in which the comparison is made between WT and mutant animals that may develop secondary and non-cell-autonomous phenotypes over time, distorting the interpretation of the process under study. However, the creation of proper models is particularly challenging since they must respond to several requisites: (a) mosaicism should be induced as early as possible during cortical formation; (b) the frequency of mosaic mutations should be tunable at will; (c) the genotype of individual neurons should be identified univocally by the expression of a fluorescent tag amenable to be imaged in vivo for anatomical and functional studies at the level of single cell and for the study of brain circuitry. Finally, this methodology should be easily transferred on the available mouse models without the need of generating mouse lines.

Focal gene expression can be induced during late embryonic age in a wide range of brain structures by in utero electroporation (IUE)[20] allowing the study of cortical circuitry in normal and pathological conditions[21–24]. To date, the generation of genetic mosaics involves the scattered activation of Cre recombinase in a floxed background and this is achieved by transfecting a low concentration of a Cre-expressing vector in order to recruit a sparse Cre-expressing cellular subpopulation[25] or by regulating the inducible Cre recombinase activation through the administration of low tamoxifen concentrations. These approaches are affected by the low reliability of conventional fluorescent Cre reporters in detecting transient or low activation of Cre recombinase, thus leading to defective reporting of the genomic recombination state[26,27]. None of these tools has been specifically devised for the generation of tunable KO/WT binary mosaics of single genes of interest during development. Most of them have a very limited control of mosaicism degree, they often suffer from false positive–negative cells readout, and in some cases, they require the use of specific animal strains preventing an easy application to a broad spectrum of possible genetic targets.

In this work, we describe the generation and characterization of Beatrix, a Cre-reporting architecture capable of amplifying and preserving weak or transient Cre events with a tenfold increase in sensitivity compared to canonical reporter structures. We demonstrate that Beatrix is a reliable reporter of Cre-mediated recombination and that it allows the creation of genetic mosaics of arbitrary degree that are amenable to be imaged in vivo by two-photon microscopy. We also show how the implementation of Beatrix to the widely diffused tamoxifen-inducible Cre system results in an enhanced sensitivity to tamoxifen mediated induction. As a proof of principle of the methodology, we create a mosaic of expression of the autism-related gene PTEN, a constitutive inhibitor of the mTOR pathway, and we describe the anatomical phenotype of this mosaic in the mouse cortex and olfactory bulb (OB). Finally, we demonstrate that the cortical mosaic is characterized by impaired network activity and by transient episodes of hyperexcitability strongly reminiscent of the electrophysiological signature of the human disease.

## Results

Our goal is the development and validation of a Cre-based tool to differentially label both WT and KO cells for a gene of interest, with a fine control of the mosaicism level. The general idea is to co-transfect two different plasmids. The first plasmid, based on the Cre-switch design[28], is a Cre reporter relying on the FLEx system[29] that switches expression between two fluorophores (RFP/GFP) depending on Cre activity. The second plasmid carries Cre recombinase and it acts as a trigger for the recombination. The concentration of the trigger plasmid determines the mosaicism degree.

**Beatrix is a reporter and effector of Cre recombinase**. Figure 1a shows the scheme of a simple reporter of Cre activity formed by the genes of a red fluorescent protein (RFP, DsRed2) and of GFP assembled in opposite directions within a double pair of heterotypic lox sites. When transfected in NIH 3T3 cells in its native form, the plasmid expresses RFP only. Upon Cre-mediated recombination, the insert rotates and the color switches to green. At low levels of Cre activity, this construct is not a faithful reporter of Cre recombination as witnessed by the presence of cells expressing both the red and green forms of the Cre-switch plasmid (Fig. 1a, lower panels). This ambiguity would inevitably

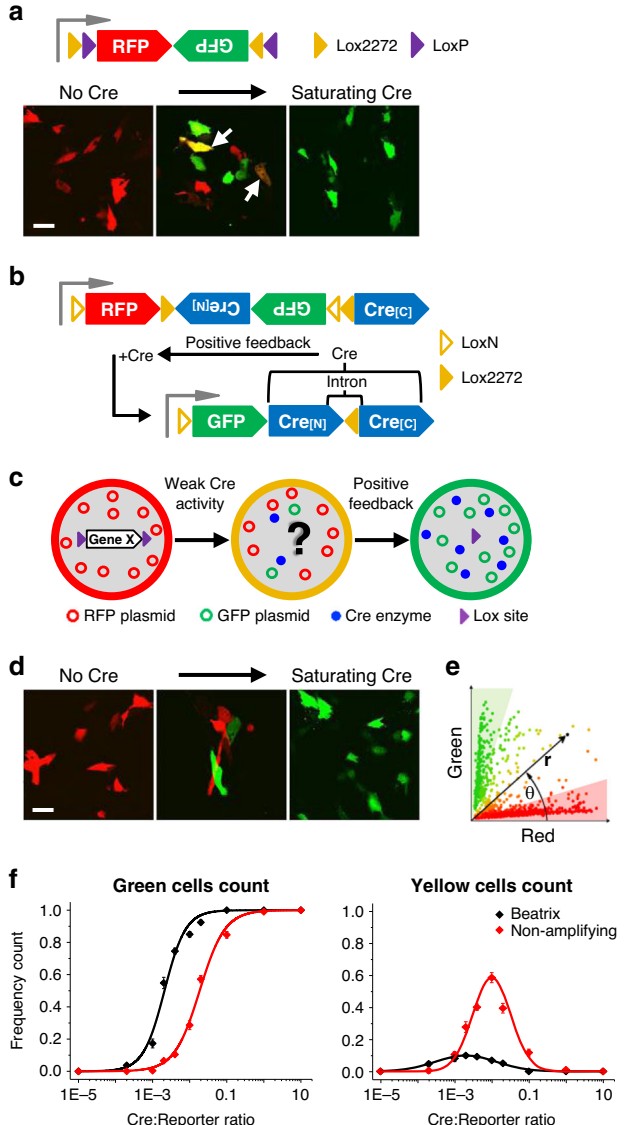

**Fig. 1 Beatrix: a sensor and effector of Cre activity. a** Schematic representation of a conventional double reporter of Cre activity based on the FLEx system (Cre-switch) and representative two-photon images of NIH3T3 cells transfected with a non-amplifying double reporter at three different ratios (No recombination: reporter only; Mosaic: 1:250 co-transfection ratio; Full-recombination: 1:1 co-transfection ratio). At low controller titer the mosaic includes a significant number of yellow cells (white arrows). Scale bar 20 μm. **b** Working principle of the Cre-embedded amplifier. See Supplementary Fig. 1 for the recombination sequence. **c** Schematic representation of the operative mechanism of the sensor. **d** NIH3T3 cells transfected with Beatrix at three different molar ratios (No recombination: reporter only; Mosaic: 1:250 co-transfection ratio; Full-recombination: 1:1 co-transfection ratio). Scale bar 20 μm. **e** Polar coordinates representation of red and green fluorescence. The radius $r$ measures the overall fluorescence intensity. The polar angle θ measures the hue of each cell, proportional to the ratio between red and green fluorescence (see Methods, Supplementary Data). Cells lying in green and red shaded areas are expressing either the fully recombined plasmid (green) or the native plasmid (red). All other cells are considered partially recombined (yellow cells). **f** The left panel shows the relative frequency of cells that express only GFP as a function of the Cre:Reporter plasmid ratio for Beatrix (black) and the non-amplifying control (red). The right panel shows the relative frequency of cells presenting mixed expression of GFP and RFP. Every point represents the mean number of green or yellow cells. Each point expresses the mean ± standard error of the mean (SEM) between all replicates; $n = 3$-4 replicates for each point, with 500–1000 cells for each replicate.

lead to false positive or negative cells when this mosaic approach is associated with a floxed conditional allele.

This experiment shows that the low titer of the trigger plasmid required to obtain an expression mosaic is not compatible with the recombination in the target cells of all the lox sites, thus leading to incomplete recombination of the sensor and to an undetermined state of the genome (Fig. 1a).

We reasoned that we might overcome this situation if the sensor was including an element that amplifies and stabilizes the initial weak recombinase activity induced by the sparse transfection of the trigger plasmid. To this effect, we developed Beatrix, an amplifier of Cre activity sensitive to low Cre concentration, which embeds a Cre-dependent Cre gene copy within a FLEx structure (Fig.1b). In absence of Cre activity, Beatrix drives the exclusive expression of RFP. Upon Cre-mediated recombination, the Beatrix cassette rearrangement leads to the RFP excision, and the expression of both GFP and Cre (Fig. 1b and Supplementary Figs. 1 and 2).

This method exploits the high plasmid copy number of Beatrix into transfected cells and its inherent positive feedback to amplify a weak Cre trigger into saturating Cre activity (Fig. 1c). Accordingly, RFP corresponds to completely absent Cre, while GFP corresponds to strong Cre expression. We initially explored

simpler variants of our construct in which the whole intron-interrupted Cre gene was inserted in antisense in the Cre-switch architecture. Unfortunately, these constructs presented a substantial leakage caused by self-activation of the embedded Cre amplifier, as recently reported[30]. We eventually determined that the key breakthrough was the division of the Cre gene in two exons assembled in opposite orientations that prevents the expression of the amplifier in absence of specific Cre-mediated recombination (Supplementary Fig. 3).

Figure 1d shows how, in presence of the amplification provided by Beatrix, NIH 3T3 cells appear clearly segregated in red and green cohorts. We quantified the difference between a conventional Cre-switch reporter and Beatrix by looking at the number of green and yellow cells over the total transfected population as a function of the Cre trigger titer, adopting the criteria depicted in Fig. 1e (see "Methods"). Figure 1f shows the frequency of cells expressing only GFP in different mosaics. The embedded amplification strongly enhances the sensitivity of Beatrix to low trigger plasmid titers, shifting the 50% point of the dose-effect sigmoidal fit toward a tenfold lower concentration (from a Cre:Beatrix ratio of $0.018 \pm 1 \times 10^{-3}$ to $0.002 \pm 1 \times 10^{-4}$) without affecting the slope. At the same time, the frequency of recombinant cells with uncertain status (yellow cells) is greatly reduced.

**Generation of expression mosaics in vivo.** Next, we tested our approach for the generation of mosaics in vivo in the mouse brain after IUE[20] of CD1 or C57BL/6J mice with Beatrix and variable concentrations of the Cre trigger plasmid. IUE was timed in order to target neurons migrating to the upper layers of the cortex (layers 2/3). Figure 2a shows a low-magnification mosaic of the craniotomy performed on a CD1 mouse at postnatal day 30 (P30) after IUE at 1:50 Cre:Beatrix molar ratio. This image results from the maximum projection obtained from 25 sections imaged every 10 μm from the surface (Fig. 2b). It is important to note that, because of the maximum

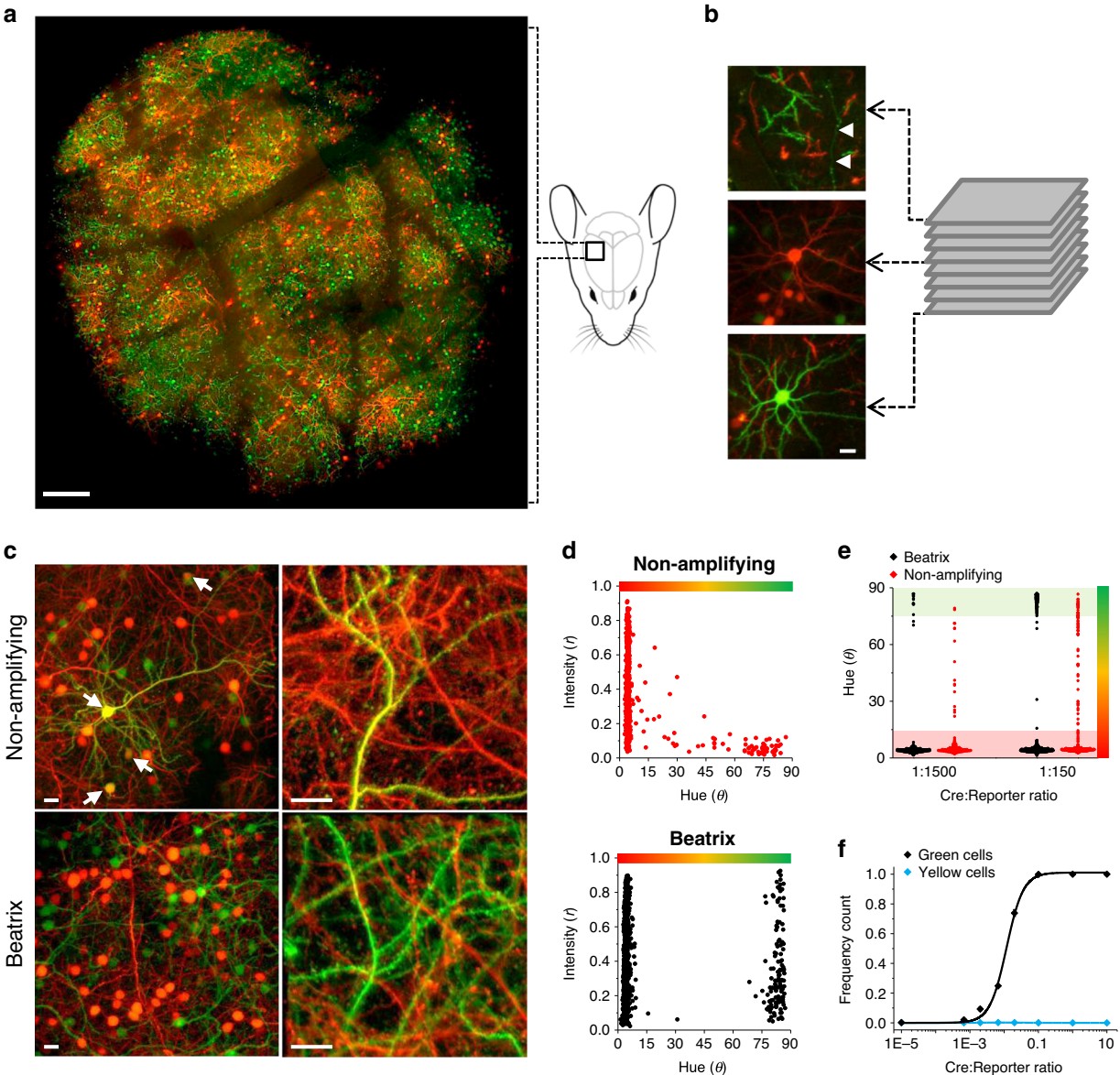

**Fig. 2 In vivo expression mosaics induced with Beatrix or with a conventional reporter. a** Wide-field mosaic showing the entire area enclosed by the cranial window. The image obtained at P30 is a composite maximum projection of a stack imaged every 10 μm from the surface down to about 250 μm depth. Scale bar 200 μm. **b** Details relative to fields placed at 25, 130, and 230 μm depth. White arrowheads point to a superficial axon in layer 1 running through a field of apical dendrites. Scale bar 20 μm. **c** Representative fields of expression mosaic obtained with a Cre:Reporter ratio of 1:150. Arrowheads indicate cell bodies expressing both RFP and GFP that are frequent in the mosaic obtained without Cre amplification. In absence of amplification, dendrites show a uniform co-localization of GFP and RFP, which is absent in presence of amplification. Scale bar 20 μm. **d** Cumulative data obtained from $n = 4$ mice transfected with Beatrix and Cre at the 1:150 ratio (864 neurons) and from $n = 3$ mice transfected with a non-amplifying reporter and Cre (609 cells); these plots show the effect of the amplification on binarization (polar angle θ) and mean fluorescence intensity (radial distance r). In absence of amplification, the strong negative correlation between intensity and hue is confirmed by Pearson correlation coefficient $R = -0.45$ (two-sided F-test of overall significance for linear regression $p < 10^{-4}$). The fluorescence intensity of the recombined cells is larger and more broadly distributed with Beatrix ($p < 0.01$, two-sided Kolmogorov-Smirnov test). **e** Scatterplot showing the difference in hue variation upon a tenfold dilution of the Cre controller using either Beatrix or the non-amplifying reporter. Cells show a very strong binarization only in presence of amplification. 1:1500 ratio ($n = 3$ mice per group), 1:150 ratio ($n = 4$, Beatrix; $n = 3$ control). **f** Frequency count of cells expressing only GFP (black) or both RFP and GFP (cyan) as a function of the Cre:Beatrix plasmid ratio. The number of yellow cells is negligible for every Cre:Beatrix plasmid ratio. Each point expresses mean ± SEM of all animals for that condition and its standard error; $n = 3-4$ animals per point, each one 300–500 acquired cells.

projection, the images of red and green cells can overlap and produce yellow patches that are not an indication of erroneous recombination. All the quantifications provided have been obtained by analyzing single optical sections (~1 μm thickness) imaged at high magnification, allowing single-cell resolution analysis (Fig. 2b). Figure 2c shows representative fields of neuronal mosaics in layer 2/3 (L2/3) of the occipital cortex

acquired in vivo at P30 after IUE with Cre and either the non-amplifying reporter (upper images) or Beatrix (lower images) at 1:150 molar ratio. We assessed these mosaics by quantifying the fluorescence signal of each neuron in the green and red channels (Fig. 2d). In absence of amplification, mosaics are affected by the presence of partially recombined yellow cell bodies and dendrites, independently of the mosaicism level (Fig. 2d, e).

Conversely, the mosaics induced by Beatrix are virtually perfectly binarized, with a negligible presence of partially recombined cells (<0.1%). Importantly, the segregation of neurons in the two classes is independent of the concentration of the triggering Cre plasmid. Thus, changes in concentration of Cre result only in a tuning of green to red cells number, without any significant increase in the yellow cells component (Fig. 2f). This result shows that different degrees of mosaicism can be achieved simply by varying the Cre:Beatrix ratios. A further notable feature of the mosaics induced by Beatrix is the high mean fluorescence intensity of the transfected cells, regardless of their recombination status. By contrast, Fig. 2d shows how the non-amplifying reporter presents a clear correlation between fluorescence intensity and hue, as fluorescence progressively decreases from red to green cells. Importantly, we verified that the constitutive expression of Cre recombinase caused by Beatrix did not affect neuronal phenotype by assessing neuronal body size, cortical layering and dendritic spine density (Supplementary Fig. 4). Beatrix can be used to create mosaics also in neuronal culture. Supplementary Fig. 5 shows the induction of a mosaic of expression of the X-linked MeCP2 gene in cultured cortical neurons obtained from a MeCP2lox background (see "Methods"). Rett syndrome is caused by mosaicism of MeCP2 in females and therefore a mosaic of expression is a better model than the commonly used homozygous KO. As expected, immunohistochemistry confirmed the MeCP2 ablation in green cells. The loss of MeCP2 caused a mild phenotype on dendritic spines: spine density and length are larger in the KO cells compared to the nearby WT neurons. Since imaging was performed at DIV12, cortical neurons are still very immature and this suggests an accelerated development of dendritic spines which is consistent with in vivo results in which a premature closure of their critical period for structural plasticity has been demonstrated[31].

**Beatrix as an amplifier of inducible Cre**. Mutations can be induced in a specific temporal window through a widely diffused inducible version of Cre recombinase whose activity can be triggered upon the administration of a specific activator (Tamoxifen)[32,33]. This inducible recombinase exploits the compartmentalization of the enzyme in the cytosolic space in absence of tamoxifen, rather than a real activation/inactivation of the catalytic function. Upon induction, the enzyme is free to move into the nucleus and to reach its substrates. The time window of Cre activation is about 4–24 h after a single intraperitoneal injection[34], and the effect decays within 36–48 h[34].

To test our tool in this inducible paradigm, we co-transfected either Beatrix or the non-amplifying reporter, as a control, with different molar ratios of the commercially available ER^{T2}-Cre-ER^{T2} plasmid (Addgene #13777).

Despite its effectiveness, the tamoxifen-inducible system is well known to suffer a certain degree of background leakage activity and, indeed, the ER^{T2}-Cre-ER^{T2} plasmid is the result of a technical refinement directed at decreasing its leakage[32,33]. For this particular version of inducible Cre, no leakage activity has been detected by adopting a conventional reporting structure[33]. However, we expected that the internal amplification provided by Beatrix would magnify any residual leakage. Indeed, Fig. 3a, b (left panels) shows how an in vivo co-transfection of ER^{T2}-Cre-ER^{T2} and Beatrix at a 1:1 rate, after 45 days from IUE produces a substantial amount of recombination with cells exhibiting a troublesome variety of hues, indicating a large heterogeneity in the recombination state.

The large spectrum of different RFP/GFP contributions at the single-cell level in the Beatrix transfected population, reflects the time course of the leakage process. Each GFP^+ cell experienced an unwanted recombination event at a different time point. This experiment shows the Beatrix capability of unmasking these transient events, turning them into triggers for the irreversible amplification process. It is critical to stress that the background activity reported by Beatrix is not an artifactual consequence of the amplification system itself, but rather an intrinsic side effect of inducible systems that cannot be properly detected by a conventional reporter[33]. This failure of detection unavoidably leads to false negative-labeled cells, depending on time, enzyme concentration and target gene locus recombination efficiency.

To solve this issue, we posed that the background recombination activity of inducible Cre is directly proportional to the enzyme concentration. Therefore, the background leakage must decrease upon lowering the ER^{T2}-Cre-ER^{T2}:Beatrix ratio. The dose-effect relationship reported in Fig. 2f shows that a full recombination of the transfected pool was already reached for the 1:10 Cre:Beatrix ratio implying that at this molar ratio the two plasmids are co-transfected in all cells. Thus, we predicted that, by transfecting neurons in utero with a 1:10 ER^{T2}-Cre-ER^{T2}:Beatrix ratio, we should significantly reduce the inducible Cre background activity, without losing the capability of reaching a full recruitment upon tamoxifen administration. Figure 3a, b (right panels) shows how in this condition there is no sign of leakage activity in the cortex 45 days after electroporation. To confirm the possibility of achieving full recruitment upon induction, we treated the animals with a single intraperitoneal injection of tamoxifen at P15, and we checked the effect 15 days after treatment. The design of this experiment is schematized in Fig. 3c. The aim was to evaluate the effect of the amplification of one transient Cre activation episode. Tamoxifen treated animals were dissected at P30 and brain slices were immunostained against Cre, to verify the effect of the amplification process. Under these conditions, mice electroporated with Beatrix show a complete recruitment over the transfected neuronal population, with every fluorescent cell found positive for the GFP labeling. Occasionally, some residual RFP fluorescence is present only in the nuclei of strongly expressing neurons because of the high stability of DsRed2 protein[35]. The prolonged intracellular lifespan of DsRed2, that can last several weeks, is attributed in a significant extent to its tetrameric structural organization and nuclear accumulation[35]. The amplification effect of the positive feedback over the transient Cre activation is witnessed by anti-Cre immunohistochemistry experiments, showing a strong Cre immunostaining signal localized into the nucleus 15 days after the induction (lower panels of Fig. 3d).

By comparison, analogue experiments performed by using the non-amplifying reporter show a substantial difference in the fluorescence profiles. A high RFP signal is still present in almost all transfected cells regardless of their fluorescence intensity. This result is caused by partial recombination of the reporter (Fig. 3d, upper panel). Consistently, anti-Cre immunostaining in these animals also show a completely different profile, with a total lack of Cre^+ cell, confirming that the recombinase activity was temporally confined to the 24–36 h following the injection[34].

Notably, whereas conventional tamoxifen administration protocols require daily tamoxifen injections delivered in the span of several days to reach maximal levels of recombination[34], Beatrix drives the complete recombination of the transfected population with just one injection, thus reducing the widely recognized harmful effects of tamoxifen toxicity. Moreover, this property would simplify the induction of complete genomic recombination when the system is applied to mice containing multiple floxed genomic sites resulting from crossing different floxed lines.

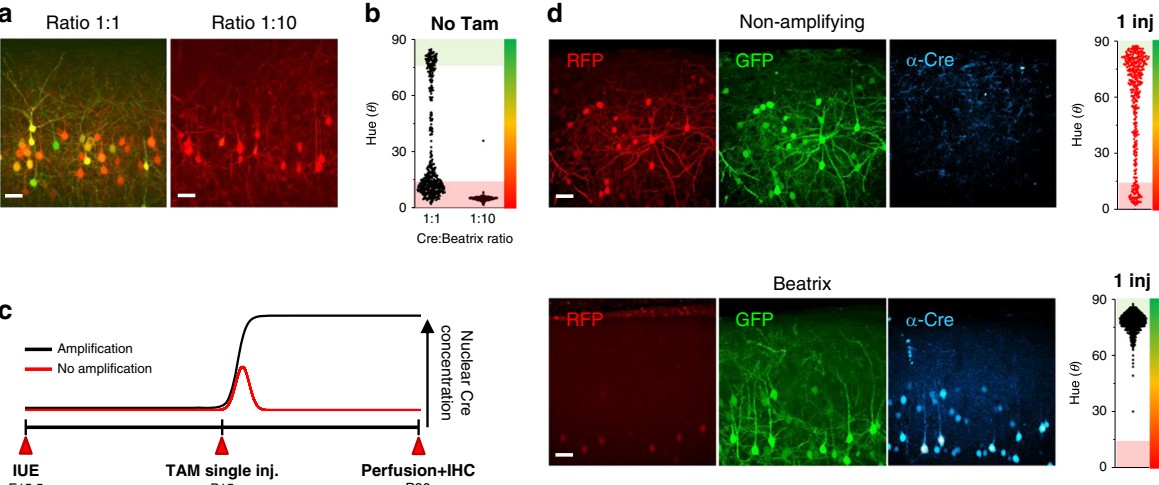

**Fig. 3 Beatrix amplifies and preserves the transient activity of tamoxifen-inducible Cre. a** Cortical neurons following in utero electroporation of ER$^{T2}$-Cre-ER$^{T2}$ together with Beatrix in a 1:1 ratio, in absence of tamoxifen at P45. The leakage accumulated in the time interval between electroporation and imaging. This effect can be countered by lowering the concentration of the ER$^{T2}$-Cre-ER$^{T2}$ plasmid to a 1:10 ratio. Scale bar 50 μm. **b** Scatterplot showing the difference in hue distribution resulting from the leakage activity of ER$^{T2}$-Cre-ER$^{T2}$ with Beatrix at the two different plasmid ratios. ($n = 3$, $n = 4$ mice; $n = 487$, $n = 315$ cells). **c** Timeline of the experiment. ER$^{T2}$-Cre-ER$^{T2}$ was electroporated in utero together with Beatrix or the non-amplifying reporter in a 1:10 ratio. At P15 we performed a single tamoxifen injection (100 μg/g animal weight) that usually does not lead to complete recombination. **d** A single tamoxifen injection at P15 led to a transient Cre activation in most cells, but, as witnessed by the co-expression of GFP and RFP, the reporter was not fully recombined, thus proving an ambiguous reporting of the recombination state ($n = 3$ mice, 408 cells). In presence of Beatrix, every cell responds to the transient Cre activation by triggering the self-amplification, thus leading to complete recombination and to the loss of the red fluorescence ($n = 5$ mice, 1040 cells). Cells with a high level of expression show some residual red fluorescence localized in the nucleus because of the long halftime of DsRed2 protein in cells. The stabilization of the transient Cre activation was confirmed by immunofluorescence staining of Cre protein, attesting the presence of the enzyme in cell nuclei 15 days after tamoxifen induction. On the other side, no Cre-positive cells were found in animals electroporated with the inducible Cre and the non-amplifying reporter, indicating how the recombinase activity in this case is temporally confined to the 24–36 h following the injection, thus leading to the massive presence of non-recombined or partially recombined cells. Scale bar 50 μm. For each experiment, cumulative scatter plots of the polar angle values ($\theta$) of each cell are reported on the right.

**Modeling genetic mosaicism in the floxed PTEN mouse.** As a proof-of-principle demonstration of Beatrix operation, we generated a sparse mosaic of expression of the PTEN gene in a conditional KO mouse strain (Pten$^{flox}$). PTEN (Phosphatase and tensin homolog) is a phosphatase that negatively regulates the PI3K/AKT/mTOR pathway exerting an important role in the regulation of several crucial cellular processes. The loss of PTEN plays a pivotal role in cancer progression, non-cancerous neoplasia, focal cortical dysplasia and has been associated with ASD, therefore providing multiple reasons of interest for mosaicism modeling[36–39]. In this set of experiments, we used Beatrix in two different contexts: in cortical pyramidal neurons after IUE and in the granule cells, the major population of inhibitory GABAergic neurons of the OB after postnatal electroporation (Figs. 4 and 5, Supplementary Figs. 6, 7). Figure 4a shows a representative field from a PTEN KO mosaic induced in the cortex with a 1:80 molar ratio that was chosen on the basis of the Beatrix dose-effect response (Fig. 2f) in order to obtain a mosaicism of the transfected neurons of about 50–60%. We estimated the fraction of mutant neurons by normalizing the number of PTEN-KO neurons (identified by the green fluorescence) with the total number of neurons identified by immunofluorescence against the neuronal marker NeuN in layer 2/3 (Supplementary Fig. 6). The mean fraction of mutant neurons was about 10% (range 6–21%, $n = 3$ mice, 16 fields, 7613 NeuN-positive neurons, 725 PTEN-KO neurons). We first focused on the validation of Beatrix as a reporter of the recombination status of the target gene, by immunofluorescence against the PTEN protein. Quantitative colocalization analysis between the immunostaining and the reporter confirms that the recombination of Beatrix is always associated with the loss of the target gene staining. Symmetrically,

all red cells are positive to the PTEN immunostaining. Together, these data demonstrate the absence of false positives and false negatives in the mosaic (Fig. 4b). Next, we studied the consequences of PTEN KO at the single-cell level. GFP$^+$ pyramidal neurons exhibit hypertrophic somata and dendrites compared to the RFP$^+$ internal controls (Fig. 4c and Supplementary Fig 7a). Previous work has shown neuronal hypertrophy of PTEN KO neurons in non-mosaic models[40,41], and our data show that the hypertrophy is cell autonomous, since the intermingled control neurons have normal size. Furthermore, the PTEN-defective pyramidal neurons presented an aberrant migratory phenotype, as shown in Fig. 4d and Supplementary Fig 7b. The KO neurons (GFP) are characterized by a larger dispersion of their depth, which cover a wider fraction of the cortex, compared to the control neurons that localize strictly in layers 2/3. Again, the defective migration is cell-autonomous and it is consistent with previous observations[42], and with the histological characteristics of focal cortical dysplasia in humans[43]. Figure 4e and Supplementary Fig. 7c, d show how the KO neurons are characterized by a strong increase in spine density, with a marked difference in primary and secondary branches and in apical dendrites, compared to the WT controls. Moreover, dendrites of PTEN-deficient neurons present marked morphological abnormalities, with a generally higher presence of immature spines (Supp. Fig. 7). The immature phenotype of dendrites is also highlighted by the high degree of residual structural plasticity observed by in vivo time-lapse imaging at P25 (Supplementary Fig 7e), an age at which the somatosensory cortex should be almost completely stabilized[44]. Notably, we also report the occasional occurrence of somatic spines in KO cells, which are not present in any of the WT somata in the acquired fields (Supplementary Fig. 7d). The

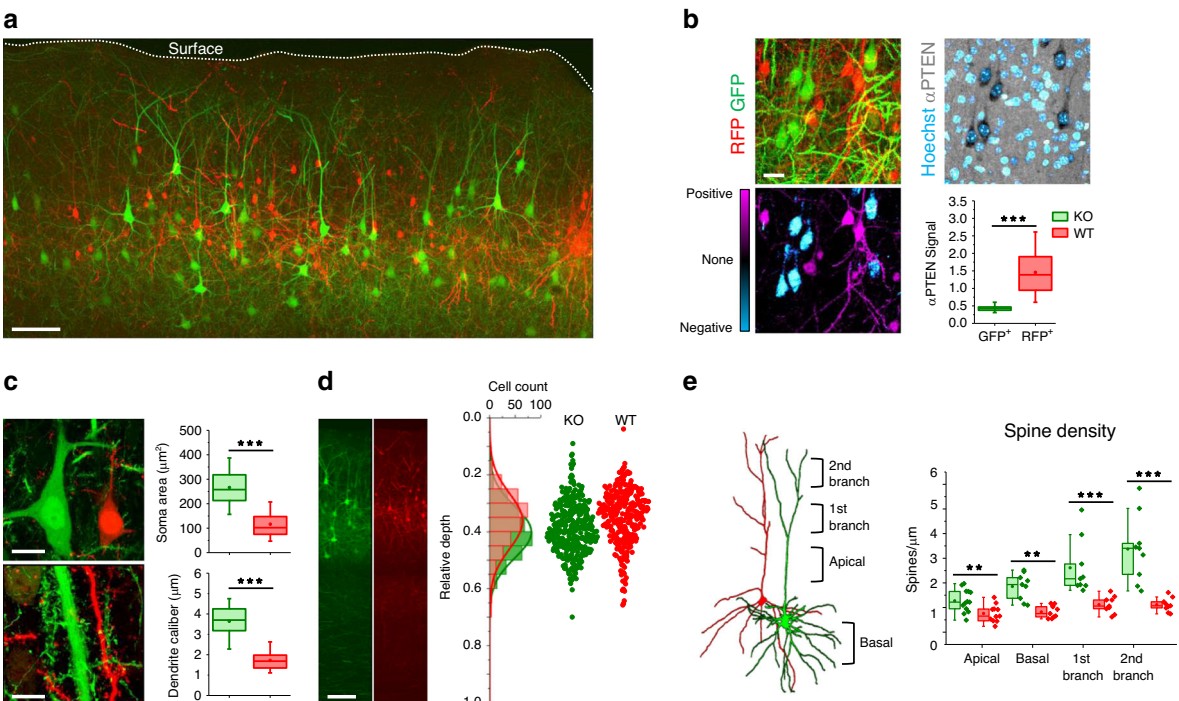

**Fig. 4 Beatrix-induced mosaic in a Ptenflox conditional mouse. a** Maximum intensity projection of a Pten^flox KO mosaic in mouse cortex. Scale bar 100 μm. **b** The lack of PTEN was confirmed by performing immunostaining against PTEN in the cortex mosaic. The colocalization analysis between Beatrix reporters and PTEN immunofluorescence highlights the absolute reliability of the system as a reporter of the genomic status. This is confirmed by the quantification of immunostaining signal in RFP/GFP positive cells (n = 3 mice; 169, 136 cells; two-sided Mann–Whitney U test, ***p < 0.001). Boxes indicate the 25 and 75 percentiles, means and medians are indicated by dots and lines respectively, whiskers indicate the 5 and 95 percentiles. Scale bar 20 μm. **c** Structural details from PTEN knockout mosaics showing that the loss of PTEN causes hypertrophy of both somata and dendrites of cortical pyramidal neurons (n = 203, 159 cells; two-sided Mann–Whitney U test, ***p < 0.001; no adjustments were made for multiple comparisons). Scale bar 10 μm. **d** The distribution of the relative depth of neurons shows that the embryonic deletion of PTEN causes a defective migration following neurogenesis, since KO cells show a larger dispersion between the cortical layers. Scale bar 100 μm. **e** Spine density was quantified in different cellular compartments, highlighting a strong increase in spinogenesis, especially in the apical compartment (n = 3 mice, two-sided Mann–Whitney U test, **p < 0.01, ***p < 0.001; no adjustments were made for multiple comparisons). In (**b**, **c**, **e**) data are represented as box plots (see the section Statistics and reproducibility in "Methods" for a detailed description).

intermingled control neurons show a normal complement of dendritic spines suggesting a strong unbalance in the excitatory input of these two neuronal population[45–47].

A similar procedure has been adopted to generate sparse PTEN KO mosaics in the OB upon postnatal electroporation (Fig. 5a). This experimental procedure labels a scattered population of inhibitory neurons in the granule cell layer of the OB. As for the IUE experiments, PTEN immunostaining confirmed the high reliability of the mosaic, showing a complete absence of false positives and false negatives within the transfected cells group (Fig. 5b). The effect of PTEN ablation on cell morphology in the OB strongly resembles the results obtained with cortical pyramidal neurons after IUE. PTEN defective neurons exhibit a hypertrophic growth of subcellular compartments, witnessed by the abnormal dimensions and shape of the cell bodies (Fig. 5c). The morphological alterations affect again also the dendritic structure, leading to an increased total length of the basal dendritic compartment (Fig. 5d). It is worth noting that PTEN defective inhibitory granule cells exhibit a strikingly different spine distribution pattern compared to the one observed in the cortical pyramidal neurons. In the basal and proximal compartments, we found a significant increase of both mature and immature spine density (Fig. 5e).

Type 2 Focal Cortical Dysplasia (T2FCD), which is the human pathology associated with PTEN somatic mosaicism, is an

important cause of intractable focal epilepsy[48,49], and it is also characterized by disturbances of sleep EEG and by the presence of characteristic 'brushes' formed by transient episodes (10–100 s) of regular firing[50,51]. In order to test whether our model is consistent with the clinical features of the disease, we compared the cortical local field potential (LFP) recorded in PTEN mosaic mice and in control mice, comparing the signal recorded in the focal mutation with the contralateral region. The recordings were performed around P30 under urethane anesthesia (Fig. 6a). In these conditions, control mice are in deep sleep and the LFP is characterized by slow oscillations from a state of low activity (down state) to transient bouts of neuronal depolarization and firing (up states). Since these transitions are finely regulated by the interaction between pyramidal cells and interneurons[52] the analysis of slow-wave oscillations (SWO) is a powerful tool for assessing the physiological state of the excitation-inhibition interaction. During slow-wave activity the entire cortex is strongly synchronized and the temporal structure of the LFP measured in the two symmetric positions is essentially identical[53] in WT mice. Surprisingly, the LFP recorded in the PTEN mosaic shows impaired synchrony with the opposite normal hemisphere. SWO are often smaller in amplitude in the mosaic and/or are of different duration (Fig. 6b). This is quantitatively demonstrated by the cross correlation between the two hemispheres computed in the PTEN mosaics and in control mice (Fig. 6c): the reduction

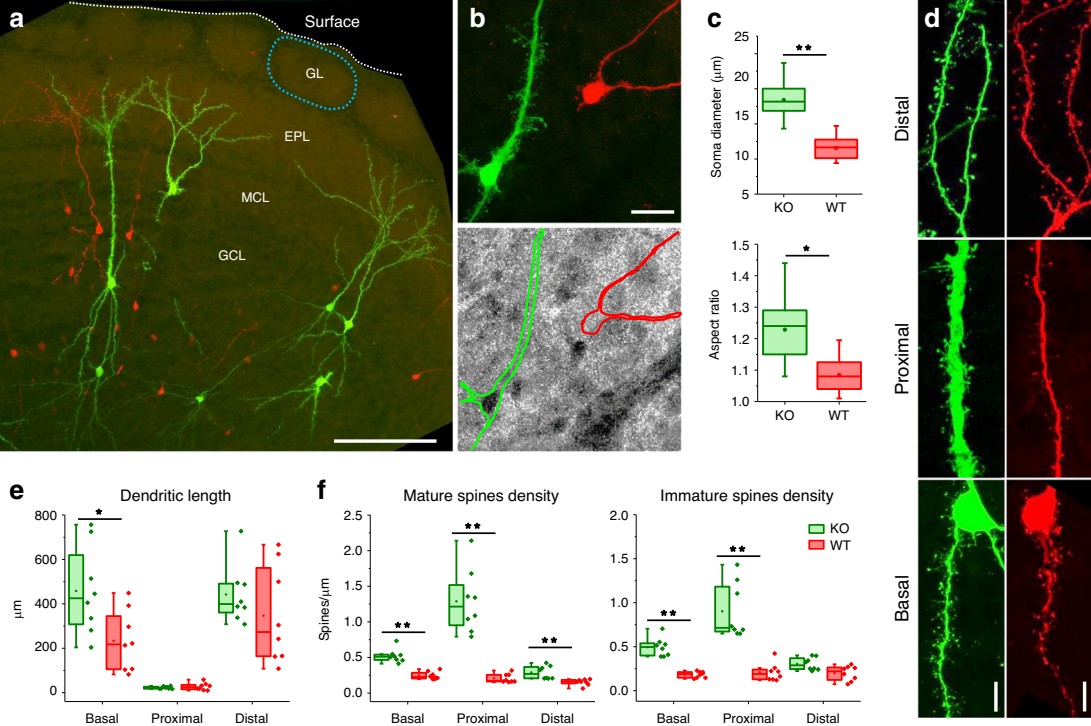

**Fig. 5 PTEN-KO mosaic of olfactory granule cells generated postnatally. a** Maximum intensity projection of a Pten^flox KO mosaic in the olfactory bulb (GL = glomerular layer; EPL = external plexiform layer; MCL = mitral cell layer; GCL = granule cell layer). Scale bar 100 μm. **b** Immunohistochemistry against PTEN confirms the loss of the protein upon Beatrix recombination (green = GFP; red = RFP; gray = PTEN). Scale bar 20 μm. **c** The loss of PTEN caused the hypertrophy of the cell body and altered its morphology. **d** Representative images of basal, proximal and distal compartments from wild type and PTEN knockout granule cells. The images show how KO neurons exhibit a general increase in spine and filopodia density, with a marked phenotype in the proximal compartment. Scalebar 10 μm. **e** The structure of the dendritic arborization is affected by the loss of PTEN as demonstrated by the length of the dendritic compartments. **f** Spine density is altered both for mature spines and for immature filopodia. Interestingly, this effect has a different spatial pattern than observed in pyramidal neurons. Statistical analysis for (**c**), (**e**) and (**f**) refers to $n = 9$ WT and $n = 9$ KO mice ($n$ spines = 1918 in WT, $n = 6205$ in KO; two-sided Mann–Whitney $U$ test, $*p < 0.05$, $**p < 0.01$; no adjustments were made for multiple comparisons). In (**c**, **e**, **f**) data are represented as box plots (see the section Statistics and reproducibility in "Methods" for a detailed description).

of cross correlation is caused by both the reduced amplitude of the up states and by their changes in temporal structure. We also computed the difference of spectral power in the δ band (0.5–4 Hz) between the two hemispheres from all our recordings. In control mice this function is centered on the zero as expected by the symmetry of SWO. In contrast, when one electrode is placed in the PTEN mosaic the symmetry is broken, the distribution becomes bimodal and strongly skewed to the left because of the lower power of oscillations in the mosaic (Fig. 6d). Interestingly, the focal mutation impairs SWO in the contralateral hemisphere: indeed, Fig. 6e shows that the δ band spectral power is lower in the control side of the PTEN mosaic mice compared to controls. A similar interference from the focus of hyperexcitability toward the opposite hemisphere has also been observed in pharmacological and genetic models of hyperexcitability[53,54]. The breakdown of the excitation/inhibition interplay is also demonstrated by the presence of two forms of epileptiform episodes recorded in 4 out of 6 PTEN mosaics. Figure 6f shows the onset of a high-frequency burst during a downstate as demonstrated by the large increase of power of the spectrogram. A second stereotyped form of activity is shown in Fig. 6g and it consists of transient quasi-periodic firing that curtails the physiological slow-wave cycle. This seizure-like activity lasted in average 32 s (range 12–60 s) and is reminiscent of the so-called brush activity recorded with intracranial electrodes in the epileptogenic dysplastic human tissue[50,55,56].

## Discussion
Herein we presented architecture for a reporter/effector of Cre activity. By implementing a regenerative amplifier of Cre recombinase, this tool allows the creation and revelation of mosaics of expression with a tunable mosaicism degree. Our tool, when electroporated in utero allows the creation of focal mosaic of expression in any available floxed mouse line, thus providing a flexible and relatively easy method to model a wide variety of neurodevelopmental disorders. The presence of a reliable fluorescent reporter of each cell genome is also useful for the study of the cell-autonomous function of a selected gene allowing the simultaneous study of arbitrary proportions of mutated and WT cells intermingled in the same environment.

In recent years, several tools have been developed for the generation and analysis of expression mosaics[57–60] with increased reporting reliability[30,57] (see Supplementary. Table 1 for a comparison). Beatrix is a tool that has been designed specifically for the study of the role of mosaicism in neurodevelopmental diseases and its main advantage respect to other techniques (e.g., MADM and MASTR)[57,58] is that it can be applied to any available floxed strain, without the need for cross-breeding with a specific reporter mouse line. In addition, our method allows a fine tuning of the mosaicism size, localization and degree, and this fine control is not achievable with other approaches aiming to generate somatic mosaic models during development (MADM).

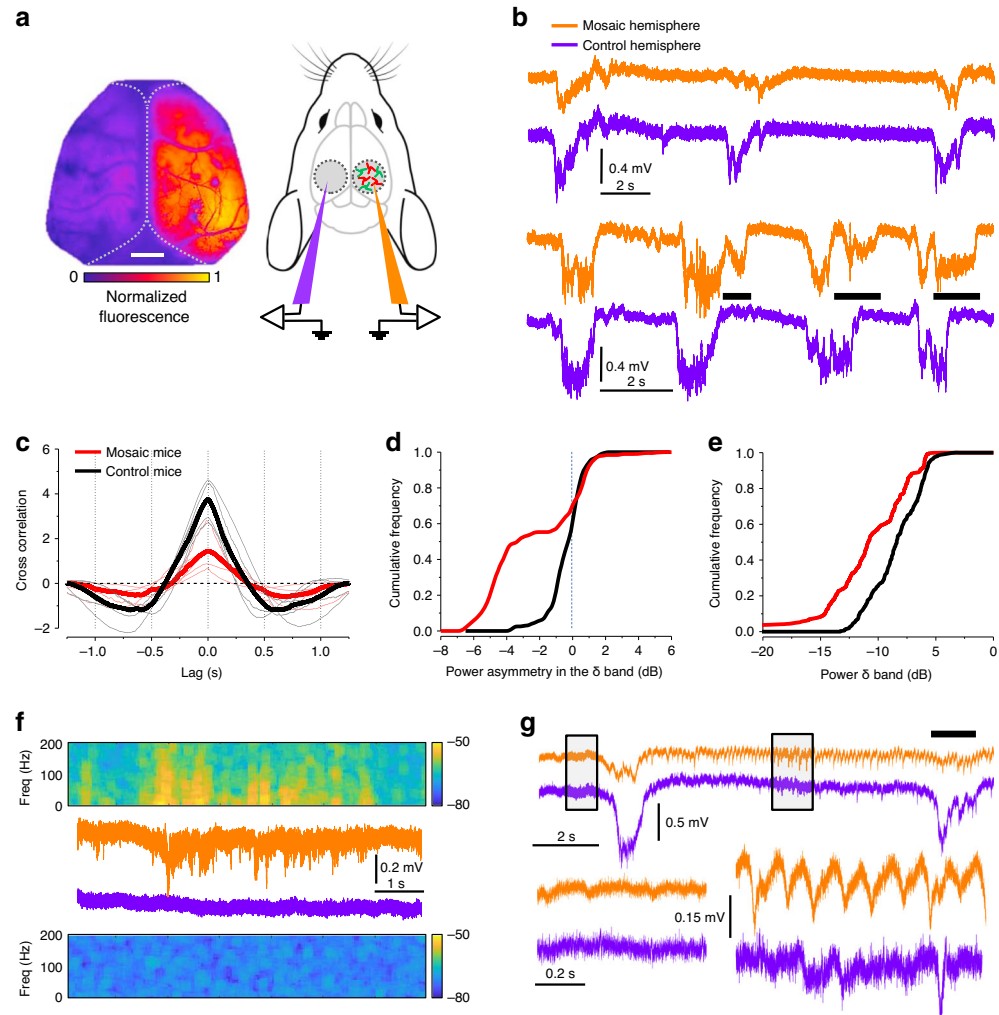

**Fig. 6 Electrophysiological impairment of the resting state EEG in the PTEN-KO mosaic. a** LFP is recorded at the center of the PTEN mosaic, as identified by wide-field imaging of Beatrix fluorescence (scale bar 1 mm), and in the corresponding site in the contralateral hemisphere in layers 2/3. **b** SWO recorded in the PTEN mosaic and in the control site in two different mice. The data show two forms of SWO anomaly: in the upper example, the PTEN mosaic exhibits a large reduction in amplitude, in the lower example, the amplitude is normal, but the timing of the oscillation is altered with long tails of activity (black bars). **c** The loss of interhemispheric synchrony is demonstrated by the cross-correlation spectra of the LFP recorded in the two hemispheres of control and PTEN mosaic mice ($n = 6$ control mice, $n = 6$ mosaic mice). Each thin line reports the spectra of an individual mouse and the thick line is the median spectra. The two groups of spectra are significantly different ($p < 0.05$, two-sided Mann–Whitney $U$ test). **d** Power difference in the delta band (0.5–4 Hz) between the two channels for the entire sample ($p < 0.01$, two-sided Kolmogorov–Smirnov test). **e** The amplitude of upstates in the control side of mosaic mice is impaired as demonstrated by the lower power measured in the δ band of individual oscillations ($p < 0.01$, two-sided Kolmogorov-Smirnov test). **f** Epileptiform burst occurring in the PTEN mosaic during a downstate. **g** Epileptiform burst formed by regular hypersynchronous spikes occurring in the PTEN mosaic. The dashed boxes indicate the areas magnified below. On the left, it can be appreciated how physiological down states are devoid of any significative activity. During the burst, however, the mosaic side exhibits periodic sharp spikes occurring at a frequency of about 5–10 Hz. These bursts are never observed in control mice. This activity suppresses regular SWO in the PTEN mosaic (black bar).

Since Beatrix involves Cre expression only in the transfected area, the risk of unwanted off-target recombination, such as in the case of other methods (MASTR and iSuRe-Cre)[30,57], is minimized. Moreover, the tight regulation of Cre amplification, coupled with the presence of a double-reporter system, allows a strict monitoring of false negatives cells, which are poorly detected in the above-mentioned systems. Recently, a method named MADR has been developed to generate in vivo mosaics exploiting both site-specific recombinase technology and IUE[61]. Although this clever system results in the generation of defined and sparse mosaics similarly to Beatrix, the MADR technique is mainly oriented to generate a gain of function mosaic rather than a KO/WT mosaic on floxed mouse strains for a specific gene of interest.

The key feature of Beatrix is that the intrinsic amplification of Cre activity endows this system with very high sensitivity in detecting weak/transient Cre levels. The advantages offered by Beatrix sensitivity are highlighted by its application to the tamoxifen-inducible Cre system, in which leakage can be reduced by employing weak promoters for Cre expression while keeping an enhanced sensitivity to tamoxifen induction. In line of principle, it should be possible to reduce the leakage also when inducible Cre is integrated into the genome, by selecting a weak promoter for Cre induction or by destabilizing its transcription. Indeed, it is known that the leakage of inducible Cre strongly depends on the Cre promoter: strong promoters (such as CaMKII in the mouse cortex) exhibit considerable leakage levels, which is drastically diminished when Cre is placed under a weaker

promoter[62,63]. The co-transfection with Beatrix would ensure the efficacy of the tamoxifen treatment also in presence of very low levels of expression of inducible Cre.

Taken together, these results confirm the robustness of the method and its extensibility to various conditions, proving Beatrix as an experimental asset to understand the physio-pathological relevance of mosaicism at the single-cell level and to extend the reporting reliability in currently available Cre-based systems, animal strains or vectors.

We exploited our tool to create a model for T2FCD based on the hyperactivation of the mTOR pathway. mTOR is an important regulator of cell growth and, in neurons, plays a crucial role in local protein synthesis, neuronal development and synaptic plasticity[64]. Accordingly, disturbances of the mTOR signaling pathway are strongly implicated in cognitive diseases and epilepsy. Since PTEN is a constitutive inhibitor of mTOR, its loss of function causes the hyperactivation of mTOR and somatic mutations of PTEN cause a variety of cortical malformations depending in severity on the extent of the mutation, ranging from hemimegalencephaly to T2FCD[43,65,66]. Several studies have exploited the PTEN conditional mouse to assess the electro-physiological phenotype consequent to mTOR hyperactivity. Most of these studies target a wide neuronal population extended in the entire brain, including the granule cells of the hippo-campus[65]. These studies have invariably found a very severe electrophysiological phenotype that includes generalized seizures. Seizures were particularly prominent in models in which PTEN ablation targeted the dentate gyrus, since these neurons are an essential regulator of the hippocampal circuitry gain[67–69]. This severe phenotype is not representative of the presentation of T2FCD in humans, in which the electrophysiological presentation is associated with sporadic epileptiform episodes[50,51,55,56].

The model provided by Beatrix has some features that well approximate T2FCD: (a) the mutation is strictly confined in a small area of the cortex; (b) it only involves genetic mosaicism in a fraction of pyramidal neurons interspersed within normal neurons; (c) the hippocampus and the majority of the cortex are not targeted by IUE and therefore have normal PTEN expression. The repetitive firing we detected shares some common features with the epileptiform activity observed in other focal models of mTOR hyperactivation[21,24,54,70] and we expect this activity to impact cognitive functions, even in absence of generalized sei-zures[71]. Finally, we found that slow-wave oscillations were strongly affected in the mosaic area and, to a lesser extent, also in the contralateral side. SWO are an essential component of the sleep cycle and any disruption of this fundamental brain rhythm is due to pose a severe cognitive burden. Indeed, slow-wave sleep facilitates the onset of epileptiform activity in epileptic patients, and specifically it has been demonstrated that sleep is a strong modulator of the pathological EEG in focal cortical dysplasia patients[51].

## Methods

**Generation of the pCAG-Beatrix plasmid**. We designed a custom plasmid (pUC-flex-GFP-P2A-Cre ex1) containing the eGFP cassette, the Lox sequences and Cre exon1. This plasmid was produced by the Genewiz gene synthesis facility (South Plainfield NJ, USA). The Cre exon2 was cloned upon amplification from a Cre ORF plasmid using BssHII and SacI sequences to obtain the pUC-flex-GFP-P2A-Cre ex1-ex2 plasmid. Then, we inserted the DsRed2 expression cassette into the AflII/XbaI sites, between the first LoxN and Lox2272 sites of pUC-flex-GFP-P2A-Cre ex1-ex2. This cloning step finally yielded the pUC-DsRed2-flex-GFP-P2A-Cre ex1-ex2 plasmid. The entire cassette was subsequently moved into an expression vector for IUE (Addgene #13777)[33] downstream the CAG promoter sequence, using EcoRI and NotI, to generate the final pCAG-Beatrix structures used in all experiments. For the generation of pCAG-BeatrixNE2 plasmid (the non-amplifying reporter that was compared to Beatrix), the Cre exon2 in pCAG-Beatrix has been removed by substituting it with an empty DNA from pUC-flex-GFP-P2A-Cre ex1, using BsrGI and SacI restriction sites. All the DNA used for cloning and transfection was produced by using the QIAGEN Plasmid Plus Maxi Kit. Trans-formations have been performed in One-Shot Stbl3 chemically competent *Escherichia coli* cell line (Thermo Fisher Scientific) to avoid unwanted recombi-nation events during plasmid preparation.

**Cell culture transfection and plasmid extraction**. HEK293T (a kind gift from Francesco Gobbo, Scuola Normale Superiorre) and NIH3T3 cells (a kind gift from Melissa Santi, CNR Nanoscience Institute) were cultured in Dulbecco Modified Eagle's Medium (DMEM) supplemented with 1 mM Sodium Pyruvate, 2 mM L-glutamine, 10 U/mL–10 μg/mL Penicillin-Streptomycin, 10% bovine fetal serum, 10 mM HEPES. All transfections were performed through electroporation on cell suspension. In brief, cells cultured on P60 Petri dishes (60 mm diameter) at 75% confluency were resuspended in 130 μl of electrolytic buffer together with 10 μg of plasmid DNA. Resuspended cells were electroporated using MicroPorator MP-100 (Digital Bio, two pulses at 1200 V with a duration of 20 ms). Imaging was per-formed on cells after 48–96 h after electroporation, because no significant alteration in red, green and yellow cells proportion was found after this time.

**Animals**. Timed pregnant outbred CD1 mice (CD-1 IGS strain code 022) were obtained from Charles River Laboratories (Wilmington, MA, USA) and main-tained at Istituto di Neuroscienze (CNR, Pisa). Inbred C57BL/6J mice (JAX stock #000664) were obtained from Jackson Laboratory (Bar Harbor, ME, USA) and reared at Istituto di Neuroscienze (CNR, Pisa). Pten^flox animals backcrossed in a C57BL6/J strain (129S4-Pten^tm1Hwu/J; JAX stock #004597) were obtained from Jackson Laboratory (Bar Harbor, ME, USA) and reared at Centro di Biomedicina Sperimentale (CBS, Pisa). CD1, C57BL/6J and Pten^flox were timed crossbred 16 days before the IUE experiment performed at E15.5. To follow the pregnancy status, females were weighed at different time points: 15, 7, and 2 days before the experiment. Animal gestational ages were then confirmed during surgery. Both male and female subjects were used for mosaics generation and analysis at different ages indicates in each specific procedure. All animal care was performed in accordance with the respective animal facilities licensing and ethical approval established in Pisa, Padua and Milan and following the Italian Ministry of Health's policies according to Dlg. 26/14. Mice were housed on a 12 light/12 dark cycle (light starting at 7AM), with temperatures of 18–23 °C, 40–60% humidity, and fed ad libitum. In particular, we followed authorized protocols #277/2015-R, #211/2020-PR and # 65E5B.N.E0K for cell culture and in vivo experiments done in Pisa, authorization #582/2016-PR for cell culture experiments in Milan on Mec-P2lox strain, and authorization # 221/2020-PR for postnatal electroporation in Padua on Pten^flox mice.

**In utero electroporation**. Tripolar IUE targeting pyramidal neurons of the visual cortex was performed as previously described[20,72] on CD1, C57BL/6J and Pten^flox mice. The day of mating (limited to 4 h in the morning) was defined as embryonic day zero (E0), and the day of birth was defined as postnatal day zero (P0). E15.5 timed-pregnant mice (gestation period ~21 days) were anesthetized with isoflurane (induction, 4%; surgery, 1%), and the uterine horns were exposed by laparotomy. The DNA (3–4 μg/μl) together with the dye Fast Green (0.3 mg/ml; Sigma, St Louis, MO, USA) was injected (~1 μl) through the uterine wall into the right lateral ventricle of each embryo by a 33-gauge standard point hypodermic needles (Sigma-Aldrich, Germany). After soaking the uterine horn with a pre-warmed phosphate-buffered saline (PBS) solution (137 mM NaCl, 2.7 mM KCl, 10 mM Na$_2$HPO$_4$, and 1.8 mM KH$_2$PO$_4$, pH 7.4, all Sigma-Aldrich), the embryo's head was carefully held between tweezers-type circular electrodes (5-mm diameter), while the third elec-trode (5.5×3.5×1 mm, platinum plate, Cooksongold) was accurately positioned on the occipital lobe. Electroporation was accomplished with a BTX ECM 830 pulse generator (BTX Harvard Apparatus) and BTX 47 tweezertrodes. Six electrical pulses (amplitude, 30 V; duration, 50 ms; intervals, 1 s) were used for electro-poration. The uterine horns were returned into the abdominal cavity, and embryos continued their normal development until delivery.

The success rate for electroporation experiments varied depending on the mouse strain. For WT C57BL/6J and CD1 animals, the survival rate after treatment was among 70 and 75%, whereas for Pten^flox animals was about 50%. In all these cases the electroporation was effective in more than 90% of the surviving treated animals.

**Postnatal electroporation**. pCAG-Beatrix plasmid electroporation targeting neuronal precursors of interneurons in the subventricular zone (SVZ) was per-formed in Padua in Pten^flox mice at postnatal day 2–4. Pups were anesthetized by placing them in a glass Petri dish pre-chilled at 4 °C and then placed in wet ice. The DNA (4 μg/μl) was injected (2 μl) with a Hamilton syringe into the lateral ventricle. The site of injection was approximately equidistant from the lambdoid suture and the eye, 2-mm lateral to the sagittal suture and 2-mm deep from the scalp surface[73]. Electrodes were placed on the lateral walls of the pup head and ten electrical square pulses (amplitude, 100 V; duration, 50 ms; intervals, 450 ms) were delivered (NEPA 21 type 2 electroporator, NEPAGENE). Electroporated pups were warmed on a heating pad before returning into the home cage. The survival rate after treatment for Pten^flox animals was about 80%. The electroporation was effective in about 80% of the surviving treated animals.

**Surgical procedures**. Experiments were performed on CD1, C57BL/6J and Pten[flox] mice (males and females) between postnatal day 28 and 32. Mice were anesthetized with an intraperitoneal injection of urethane (1.6 g/Kg, injected as 20% w/v solution in physiological saline). During the experiment the animal respiration was aided providing $O_2$ enriched air and body temperature monitored and held constant at 37 °C with a feedback-controlled heating blanket (Harvard Instruments). The animal head was shaved and 2.5% lidocaine gel applied to the scalp. Scissors were used to cut the flap of skin covering the skull of both hemispheres; the exposed bone was washed with saline and periosteum was gently removed with a pair of forceps to provide a better adhesion for glue and dental cement. The area expressing the highest fluorescence was identified with transcranial epifluorescence illumination. A custom-made steel head post with a central imaging chamber was then glued with cyanoacrylate in a plane approximately parallel with the skull over the cortical region of interest and cemented in place with white dental cement (Paladur). The mouse was head-fixed and a craniotomy of 2–3 mm in diameter was drilled over the region of interest; care was taken to minimize heating of the cortex during surgery, dural tears, or bleeding, and to keep the cortex superfused with sterile ACSF (126 mM NaCl, 3 mM KCl, 1.2 mM $KH_2PO_4$, 1.3 mM $MgSO_4$, 26 mM $NaHCO_3$, 2.4 mM $CaCl_2$, 15 mM Glucose, 1.2 mM HEPES in deionized $H_2O$, pH 7.4).

**In vivo two-photon image acquisition**. Fluorescence was imaged with a Prairie Ultima Multiphoton microscope (Bruker) and a mode-locked Ti: sapphire laser (Chameleon Ultra II, Coherent) through a 20× Olympus XLUMPLFLN water immersion objective (numerical aperture 1.0). Before each imaging session we measured the power of the excitation laser at the optic bench and at the output of the objective lens at each wavelength. From these data, we computed the power at the sample, which is not accessible once the mouse is placed under the objective, in order to maintain it under 30 mW. Fluorescent cells were imaged at variable depth in the cortex. All two-photon imaging experiments have been performed at the excitation wavelength of 950 nm. The emission filters band-pass used were 490–560 nm for the green channel and 584–680 nm for the red channel. For cultured cells experiments, multiple snapshots at a resolution of 512 × 512 or 1024 × 1024 pixels and zoom 1, were taken 48 h after cell transfection. In vivo imaging was performed by acquiring a set of Z-stacks in the transfected regions. Dark frames were acquired after closing the laser shutter to measure the mean noise arising in the PMTs and the pedestal usually added by the electronics. Time-lapse imaging of dendritic structures was restricted to the apical dendrites present in cortical layers 2/3 (50–150 mm below the cortical surface) and was conducted for at least 50–60 min at an interval of 5 minutes. Images were acquired at a resolution of 512 × 512 pixels at zoom 10, leading to a field of about 50 μm and a nominal linear resolution of about 0.1 μm/pixel. The stack step size was 0.75 mm. The acquisition of each frame was synchronized to the heartbeat and this procedure strongly reduced the mechanical artefact produced by the pulsing brain circulation and improved the alignment of the image stacks[74]. Analysis of the imaging data was carried out using ImageJ. The optical sections of each stack were aligned to compensate for mechanical movements by using the SIFT ImageJ plugin[75].

**Cell culture and transfection of primary mouse cortical neurons**. Cultured mouse cortical neurons were prepared from E16.5 embryos of WT or Mecp2lox mice[72] maintained on a C57BL6/J background (MMRRC; B6; 129S4-Mecp2tm1Jae/Mmucd; stock number: 011918-UCD) in Milan. Neurons were plated at 150–200 cells/mm² density on 12-well plates (Euroclone) coated with 0.01 mg/ml poly-L-Lysine (Sigma-Aldrich) and maintained in Neurobasal (ThermoFisher) supplemented with B27 (ThermoFisher). At day-in-vitro 7 (DIV7), neurons were transfected using Lipofectamine 2000 (ThermoFisher). Imaging experiments were performed at DIV14.

**Immunocytochemistry**. Cells were fixed in 4% paraformaldehyde, 4% sucrose in PBS at room temperature for 10 min. Rabbit αMeCP2 (Cell Signaling, #3456) primary antibodies were dissolved 1:500 in home-made gelatin detergent buffer (GDB, 30 mM phosphate buffer, pH 7.4, 0.2% gelatin, 0.5% Triton X-100, 0.8 M NaCl, all Sigma-Aldrich) and applied overnight at 4 °C. Secondary antibodies conjugated with fluorophores (Jackson ImmunoResearch Laboratories, code 111-175-144) were also dissolved 1:300 in GDB buffer and incubated for one hour. DAPI-stained (ThermoFisher) cells on glass coverslips were then mounted on glass slides using Mowiol mounting medium.

**Image acquisition and processing of cultured neurons**. Confocal images were obtained using LSM 510 Meta confocal microscope (Carl Zeiss, gift from Fondazione Monzino) with Zeiss 63× or 25× objectives at a resolution of 1024 × 1024 pixels. Images represent averaged intensity Z-series projections of 2–7 individual images taken at depth intervals of around 0.45 μm. For dendritic spine analysis, morphometric measurements were performed using Fiji/ImageJ software. Individual dendrites were selected randomly, and their spines were traced manually. The maximum length and head width of each spine were measured and archived automatically.

**Immunohistochemistry**. Animals were deeply anesthetized with urethane and perfused transcardially with 4% paraformaldehyde/PBS (4% PFA). Samples were post fixed overnight in 4% PFA. For immunofluorescence, brains were sectioned at 50 μm (100 μm for OB sections) thickness on a vibratome (Leica VT1000 S). Sections were processed as free-floating sections and stained with 1:100 αCre (Sigma-Aldrich, C7988), 1:350 αNeuN (Sigma-Aldrich, MAB377) and 1:100 αPTEN (Cell Signaling, #9559) antibodies. Secondary immunostainings were performed using 1:300 Alexa Fluor conjugated secondary antibodies (Abcam AF405, AF647).

**Confocal image acquisition of brain slices**. Fixed tissue was imaged with the Zeiss LSM-800 Airyscan confocal microscope with 405/488/561/640 nm lasers according to the secondary antibody. Low-magnification images were acquired using a 20× air objective (NA 0.5). Representative fields from thick slices (50 μm) were imaged by acquiring a set of Z-stacks (50 μm thick) for each experimental condition. The Z-stacks were collapsed with a maximum intensity projection to a 2D representation. Similarly, high-magnification images were acquired with a 60X oil immersion objective (NA 1.3) and post-processed to extract the details of the cellular morphology (dendrites and spines).

**Two-photon imaging analysis**. Two-photon data were composed of 12-bit images and have been analyzed with ImageJ (version 1.52t, USA National Institute of Health). For cell culture, after background subtraction, images were analyzed using a custom-made ImageJ macro function to automatically identify regions of interest (ROIs) around single cells. The macro computes the average fluorescence of each cell. In vivo Z-series were projected (maximum intensity) and corrected for dark background. ROIs were manually drawn around cellular somata: all pixels within each ROI were averaged to give the fluorescence value. Imaging data were not corrected for bleed-through of emission in the two emission channels. Polar angle and radial distance were calculated from the colorimetric plots.

Confocal acquisitions were composed of 16-bit images and were analyzed with ImageJ. Z-series for colocalization experiments, were performed through a frame by frame pixel intensity spatial correlation analysis with the Coloc2 ImageJ plugin function. For anti-colocalization experiments, the same procedure has been applied between RFP/GFP positive pixels and 0 value pixels from the AF647 Alexa Fluor conjugated immunostaining, obtained upon background subtraction. For display purposes, all images presented in the study received an identical non-linear stretching to improve visibility of the fainter details. All microscopy images are displayed with scaled color (between the minimum and maximum brightness values of the image) and with 0.5 gamma correction to enhance contrast. These corrections were used purely for representation, whereas all quantitative brightness analyses were performed on uncorrected raw images.

**Electrophysiology**. The LFP was recorded simultaneously from the injected and control hemispheres for approximately one hour after the end of the surgery under urethane anesthesia[53]. All experiments were performed around mid-day. Non-electroporated littermates were used as controls. Glass micropipettes filled with ACSF solution (impedance ~2 MΩ) were positioned into the visual cortex (2.5 mm laterally and 0.5 mm anteriorly respect to lambda) at a depth of 250–300 μm (layer 2/3). A common reference Ag-AgCl electrode was placed on the cortical surface. The potential was amplified 1000-fold (EXT-02F, NPI) by the head stage, band pass filtered (0.1–1000 Hz), and sampled at 10 kHz with 16-bit precision by a National Instruments (NI-usb6251) AD board controlled by custom-made LabView software (Labview version 13.2). Line frequency 50 Hz noise was removed by means of a linear noise eliminator (Humbug, Quest Scientific). All data were analyzed with custom-made Matlab code (Matlab 2020a, Mathworks).

**Statistics and reproducibility**. Where appropriate, data have been represented as box plots. In box plots, lower and upper bounds of the box represent the first and third quartile respectively, means and medians are indicated by dots and lines respectively, whiskers indicate the minimum and maximum values, not further than 1.5 times the interquartile range.

All representative experiments that have been shown in each figure have been quantified in other panels of the same figure and the number of repetitions or mice is indicated in the figure legends. Figure 1a, d: red, green and yellow cell ratio has been quantified in Fig. 1f; Fig. 2a–c: red, green and yellow cell presence in vivo is quantified in Fig. 2d–f; Fig. 3a: cell hue is quantified in Fig. 3b; Fig. 3d left panel: cell hue is quantified in the right panel, the result of immunohistochemistry was confirmed in all mice ($n = 3$ for non-amplifying control and $n = 5$ for Beatrix); Fig. 5a, b: parameters describing dendrites are quantified in Fig. 5c, e, f; Fig. 6a: the difference in fluorescence between mosaic and control hemispheres that is shown has been confirmed in all 6 mosaic mice that contributed to data in Fig. 6c–e.

No collected data were excluded from the analysis and repetitions of all experiments have been included in the figures, except cases of complete failure of transfection or electroporation, since no fluorescent cells were present and no data could be collected.

**Reporting summary**. Further information on research design is available in the Nature Research Reporting Summary linked to this article.

## Data availability

Source data are provided with this paper. The complete datasets generated during and/or analyzed during the current study are available from the corresponding author on reasonable request. Any other relevant data and materials are available from the author upon reasonable request. Source data are provided with this paper.

## Code availability

Matlab code for the analysis of electrophysiology recordings is available from https://github.com/GabNar/Trovato_et_al_2020. All other code specifically designed for this study is available on request.

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

## Acknowledgements

We would like to thank Silvia Burchielli, Cecilia Ciampi, Sara Ciampi and Francesca Biondi for their technical help in animal care. The study was supported by Telethon grant GP19281 to CL and GMR and by Regione Toscana project DECODE-EE and by PRIN 2017-20175C22WM. We are grateful to Renzo Guerrini for comments and to Pasqualantonio Pingue, Luigi Rolandi, Lucia Sorba and Michela Matteoli for facilitating experimental work during the COVID19 lockdown.

## Author contributions

The study was designed by F.T., R.P., G.M.R. and was supervised by F.T. and G.M.R. Beatrix was designed and generated by F.T. and R.P. with the help of L.M. Early versions of Beatrix have been extensively tested in vitro and in vivo by L.C., A.C., C.L., C.S., C.V. and L.G. In utero electroporation of wild type animals was performed by L.C., A.C., F.T., O.C., S.L., E.P. and G.N. Experiments on PTEN strain were performed by F.T., E.P., S.L., O.C., G.N., F.C. and V.P. Experiments on cultures of MeCP2-KO mice were performed by C.S., C.V. and L.G. Experiments on the olfactory bulb of PTEN mice were performed by C.L. and A.M. and F.C. helped with data acquisition. The paper was written by F.T., G. M.R., E.P., O.C. and G.N. with further contributions from all authors.

## Competing interests

The authors declare no competing interests.
