## [Peer Review File · Nature Communications]

Reviewers' Comments:

Reviewer #1:

Remarks to the Author:

Title: Modelling genetic mosaicism of neurodevelopmental disorders in vivo by a Cre-amplifying fluorescent reporter

Recently, it has been known that brain somatic mosaicism plays a key role in various neurological disorders including focal epilepsies. Thus, in vivo modelling of brain somatic mosaicism is important to understand the molecular genetic pathogenesis of these conditions. The authors generated and characterized Beatrix, a new Cre reporting architecture capable of amplifying and preserving weak or transient Cre events, which is a reliable reporter of Cre-mediated recombination and allows the efficient modelling of brain somatic mosaicism. However, this system did NOT show any behavioral defects at in vivo level (e.g. spontaneous behavioral seizures), compared to previous models. Also, several concerns need to be addressed by the authors to provide a sufficiently significant conceptual advance to merit publication in Nature communications.

Major

1. Throughout the manuscript, the authors emphasized that the frequency of mosaic mutations should be tunable at will and the Beatrix allows fine tuning of the mosaicism size, localization, and degree. They also performed in utero electroporation with Beatrix at various molar ratio. With this regard, it is important for authors that they should show how much range of the frequency of mosaic mutations could be efficiently modelled by Beatrix. To strengthen the advantage of Beatrix for modelling brain somatic mosaicism with tunable degree, they need to resolve this issue, quantifying the number of GFP+ (recombined) neurons in electroporated mice brain with various molar ratio of Cre:Beatrix. Also, the author should show behavioral defects at in vivo level (e.g. spontaneous behavioral seizures).
2. In Fig. 2d, they counted frequency of Green cells and Yellow cells in various Cre:Reporter ratio groups. Beatrix shows the frequency of Yellow cells at nearby 0. However, in Fig 2a, a representative image by in vivo two photon microscopy with electroporated mice brain at 1:50 molar ratio, we can see considerable distinct Yellow neurons. This is an important issue since authors emphasized that Beatrix shows a reliable fluorescence reporter of each cell genome.
3. In this study, the authors developed a novel technique for efficient modelling of brain somatic mosaicism. With this regard, we thought that authors spent lots of space for explaining somatic mosaicism in several disorders including X-linked disorders, focal cortical dysplasia, autism, and etc. in Introduction part, rather than description on previous techniques for modelling somatic mosaicism such as MADR and their pros/cons, which are described only in Discussion part. Overall rearrangement of contents in Introduction and Discussion part is needed for readers to easily get the novelty of their technique.
4. In Fig1, how long did it take to capture images after co-transfection? Do they have enough incubation time for cre-expression? How do the green or yellow cell counts change over time? What is the full structure of non-amplifying reporter system? They had pCAG promoters? It is needed to describe the method section in detail.
5. In Fig. 4, the authors need to perform control experiments. For example, if they want to show that Beatrix is a better reporter system than previous models, they need to perform in vivo co-transfection with non-amplifying reporter such as pCAG-empty-reporter plasmid.

Minor

1. In Fig 1a and d, they performed experiments at three different concentrations of Cre. Exact concentration in each condition needs to be described.
2. Figure legend is too long overall. It would be better if some description in legend moves to main text.
3. Please, add scale bar in all figure images.
4. The authors need to perform some quantification and significance analyses. In suppl Fig. 3, please quantify the frequency of GFP positive cells occurred by leakage event. In line 159, 160, authors described 'the frequency of recombinant cells with uncertain status is 'massively' reduced'. They need to show its significance. Likewise, in line 172, 'considerable number' needs to be corrected specifically. Also, in line 178, 179, quantitative analysis of mean fluorescence intensity needs to be added.
5. There are some typos. In line 66, cognitive disorders ad epilepsy -> cognitive disorders and epilepsy. In Fig. 2d legend, maybe 'black symbols on bottom' and 'Red symbols on top'. In suppl Fig. 4 legend, there is no suppl Fig. 4g!
6. In line 187-198, explanation on tamoxifen inducible Cre system is too long. Maybe authors can cut a long story short.
7. In Fig 3d, authors described that no Cre⁺ cells were found in non-amplifying group. However, as they explained well about tamoxifen-inducible Cre in line 187-198, this Cre variant is localized in cytoplasm in absence of the activator, remaining its catalytic activity. So, in Non-amplifying group in Fig. 3d, we thought electroporated neurons should exhibit cytoplasmic Cre signal at least.
8. In line 270, rationale for why authors use 1:80 molar ratio needs to be added.
9. In line 286, explanation on Supple Fig 5 b,c is came up after Supple Fig 5d. Supple Fig. 5 needs to be rearranged.
10. In line 290 and suppl Fig. 5, adding some distinct markers pointing morphological abnormalities would be helpful.
11. In Fig 5b, gray colored PTEN signal looks like stronger in KO neuron rather than that of WT. Colocalization analysis and quantification of immunostaining signal need to be added like Fig. 4b.
12. In line 311, authors described that mutated granule cells in Fig.5 also presented somatic spines. Additional magnified images showing the somatic spines would be helpful like suppl Fig. 5b.
13. In line 313-320, explanation on MeCP2 gene and Supple Fig. 6 is insufficient. Also, in suppl Fig. 6, it would be better if suppl Fig. 6d moves in front of other figures, considering the context.
14. In line 167, it was written that 'Images are maximum projections obtained from 15 sections imaged every 10um from the surface (Fig 2b).' However, considering the context, the number of sections needs to be corrected to 25, not 15.
15. In line 252, they need to described the data or cite the reference; the recombinase activity was temporally confined to the 24-36h following the injection.

Reviewer #2:

Remarks to the Author:

The authors describe a modification to the Cre-dependent reporter systems commonly used for marking specific populations of cells in vivo. Specifically, they have altered a reporter that switches from RFP to GFP expression upon Cre-dependent inversion of the reporter cassette orientation to give additional expression of the Cre gene from the reporter following this inversion. This additional expression of Cre from the reporter amplifies the Cre recombinase activity in that cell ensures complete recombination of all of the reporter alleles within that cell such that a single cell exhibits only RFP or GFP expression. Standard red/green reporters can exhibit mixtures of red and green if the Cre activity in a cell is too low to recombine all of the reporter alleles in that cell. This is valuable because it reduces ambiguity in the analysis of mosaic populations where it is desirable to assess the phenotype of two distinct populations of cells.

Specifically, the new reporter construct, called Beatrix, splits the Cre gene into two fragments that, after recombination, are brought together (separated by an intron) to allow expression. This strategy reduced background Cre expression from including an inverted full-length Cre gene in the reporter.

In this manuscript, the authors describe the generation and testing of the Beatrix reporter, and show that it is useful for phenotypic analysis of two distinct neuron populations in two different brain regions of a Pten KO mosaic mouse model. They show that this strategy results in dramatically fewer "yellow" cells with co-expression of both reporter markers, as clearly shown in panels 1e, 1f for cell culture experiments, and in panels 2d-f for in vivo neuron labeling in mouse brain. The modified reporter also works well with tamoxifen-inducible Cre systems, dramatically boosting Cre expression after a single administration of tamoxifen. I'm not sure how interesting the biology of the Pten mosaic model they demonstrate in fig. 4 (cortex) and fig. 5 (olfactory bulb) is (because it's not my field), but they clearly show that they can carry out detailed phenotypic analysis of the two mosaic populations of neurons marked in vivo with Beatrix.

The Cre reporter engineering represents a novel approach that will likely be a generally useful tool for many neuroscience applications. The experiments are well-conceived, the data are clearly presented, and the manuscript is well-written. I recommend publication of this work in Nature Communications.

One possible concern is the use of green and red color schemes in all of the figures. My understanding is that a significant fraction of people are red/green colorblind and would have difficulty interpreting many of the figure panels with the current color scheme. Although I appreciate that the reporters have green and red fluorescence, the figures could easily be pseudocolored with a different color scheme (for example green/magenta) to make them more accessible.

Reviewer #3:

None

Dear Editor,

We have really appreciated the opportunity of resubmitting our study and to respond to the Reviewer's observations. Here follows a point by point response to the Reviewer comments.

Reviewer #1

Recently, it has been known that brain somatic mosaicism plays a key role in various neurological disorders including focal epilepsies. Thus, in vivo modelling of brain somatic mosaicism is important to understand the molecular genetic pathogenesis of these conditions. The authors generated and characterized Beatrix, a new Cre reporting architecture capable of amplifying and preserving weak or transient Cre events, which is a reliable reporter of Cre-mediated recombination and allows the efficient modelling of brain somatic mosaicism. However, this system did NOT show any behavioral defects at in vivo level (e.g. spontaneous behavioral seizures), compared to previous models. Also, several concerns need to be addressed by the authors to provide a sufficiently significant conceptual advance to merit publication in Nature communications.

We appreciate that the Reviewer found that Beatrix provided a reliable detection of Cre-mediated recombination. The development of this tool, its characterization and the extension of its use to inducible systems proved to be rather complex and the generation of the PTEN mosaic of expression was intended as a proof of principle of the method. However, we agree that the presentation of some features of physio-pathological relevance would increase the study interest, especially if they would share common traits to the neurophysiology of the human disease.

Accordingly, we have now performed a set of electrophysiological experiments on 6 PTEN mosaic mice and 6 matched controls under urethane anaesthesia. In these experiments we have recorded the local field potential (LFP) at the center of the Beatrix fluorescent patch and at the symmetric location on the opposite hemisphere: in this way we can directly compare the effect of the focal PTEN deletion with a matched control. This new set of results is now analysed in the new **Figure 6**.

Coherently with the fact that type 2 focal cortical dysplasia in patients is not characterized by frequent seizures, but by subtle EEG signs (Guerrini *et al.* Epilepsia, 2015), we did not observed generalized seizures but two anomalies of great patho-physiological significance: 1) disruption of slow wave activity measured during sleep and 2) the appearance of burst of epileptiform activity strongly resembling the events recorded in human patients, both during activity and sleep (Guerrini *et al.* Epilepsia, 2015; Menezes Cordeiro *et al.* Epilepsy Res. 2015).

The new material has been described in lines **312-334**. These data have been discussed in the context of the clinical signature of the disease in lines **385-412**.

The complete mechanistic understanding of these alteration is outside of the scope of this study and will require a separate full-length paper. I hope that the Reviewer will accept this as a meaningful proof of principle demonstration of the physiological relevance of the model.

Major comments

1. Throughout the manuscript, the authors emphasized that the frequency of mosaic mutations should be tunable at will and the Beatrix allows fine tuning of the mosaicism size, localization, and degree. They also performed in utero electroporation with Beatrix at various molar ratio. With this regard, it is important for authors that they should show how much range of the frequency of mosaic mutations could be efficiently modelled by Beatrix. To strengthen the advantage of Beatrix for modelling brain somatic mosaicism with tunable degree, they need to resolve this issue, quantifying the number of GFP+ (recombined) neurons in electroporated mice brain with various molar ratio of Cre:Beatrix.

We agree that this point is very important. Indeed, figure 2f shows the percentage of mosaicism obtained *in vivo* at different doses of the Cre plasmid upon *in utero* electroporation. We regret that the text was probably not explicit enough and this must have induced the Reviewer to overlook this result. We have now modified the text (see lines **180-181**) in order to provide a more direct and explicit discussion of this rather essential point.

Also, the author should show behavioral defects at in vivo level (e.g. spontaneous behavioral seizures).

We agree. See our response to the Reviewer's general comments.

2. In Fig. 2d, they counted frequency of Green cells and Yellow cells in various Cre:Reporter ratio groups. Beatrix shows the frequency of Yellow cells at nearby 0. However, in Fig 2a, a representative image by in vivo two photon microscopy with electroporated mice brain at 1:50 molar ratio, we can see considerable distinct Yellow neurons. This is an important issue since authors emphasized that Beatrix shows a reliable fluorescence reporter of each cell genome.

We thank the Reviewer for this observation because he correctly noted a potentially misleading feature of this image. The mosaic shown in figure 2a is a low magnification image obtained by performing the maximum projection of 26 focal planes from the brain surface down the depth of about 0.25 mm. Because of the projection of several different focal planes, inevitably there are several areas with a clear overlap of green and red pixels that belong to different cells placed at different depths. A second issue with this image is that some cells and processes have a very high expression of the either GFP or RFP leading to saturation of the relative channel. The bleed through of the signals leads to an apparent yellowing effect in the areas of saturated green or red signal.

This image was presented to show the extent of the mosaic rather than to provide a solid base for the quantifications that were performed on higher magnification images of thin optical sections as described in Fig 2b. We regret not to have clearly addressed this issue in the text, and we modified it accordingly in lines **164-169**. Additionally, if it is considered of sufficient importance by the Reviewer, we would be happy to add a supplementary figure with the original data showing the factors leading to the yellow areas of the image.

3. In this study, the authors developed a novel technique for efficient modelling of brain somatic mosaicism. With this regard, we thought that authors spent lots of space for explaining somatic mosaicism in several disorders including X-linked disorders, focal cortical dysplasia, autism, and etc. in Introduction part, rather than description on previous techniques for

modelling somatic mosaicism such as MADR and their pros/cons, which are described only in Discussion part. Overall rearrangement of contents in Introduction and Discussion part is needed for readers to easily get the novelty of their technique.

The last few years has seen the publication of several excellent methods for the generation of genetic mosaicism. Even if these methods share common goals, they differ in important details. Beatrix has been specifically designed for the generation of mosaic models of neurodevelopmental diseases and this is a novel feature of our method. As directed by the Reviewer we have modified the text adding a more complete outline of the existing methods (**see lines 353-384**) and we have also added a table (**Supp. Table 1**) that compares several important features of the existing methods.

4. In Fig1, how long did it take to capture images after co-transfection? Do they have enough incubation time for cre-expression? How do the green or yellow cell counts change over time? What is the full structure of non-amplifying reporter system? They had pCAG promoters? It is needed to describe the method section in detail.

We are very sorry that these data were missing from both the main text and legend. The data shown in Fig. 1 are referred to 4 days after transfection, but cells included in this part of the study have been analysed at different time points ranging from 2 to 7 days after transfection with identical results. Therefore, we believe that Cre expression was complete in this time frame. The yellow cells, due to incomplete recombination, lasts throughout this time window and, more importantly, were found also after *in utero* electroporation at P30. We have modified the methods section to add these important details in lines **440-442**.

Both Beatrix and the non-amplifying constructs are CAG driven, and they are the exact same plasmid except for the absence of the exon 2 of Cre recombinase in the non-amplifying reporter (266/343 aa of the Cre protein sequence, including the catalytic domain). Of course, we will provide the full sequences of the generated plasmids. These details are described in lines **419-434**.

5. In Fig. 4, the authors need to perform control experiments. For example, if they want to show that Beatrix is a better reporter system than previous models, they need to perform in vivo co-transfection with non-amplifying reporter such as pCAG-empty-reporter plasmid.

These experiments would be really time consuming and ethically debatable since the PTEN mice have a low rate of reproduction and the *in-utero* electroporation on this mouse line has a mediocre success rate. For this reason, we would need to electroporate several batches of mice to reach a meaningful conclusion. Figure 2 demonstrates that the recombination state of the sensor is completely undetermined in mice transfected with the non-amplifying system and the situation is not going to improve on the PTEN mice.

Minor points

1. In Fig 1a and d, they performed experiments at three different concentrations of Cre. Exact concentration in each condition needs to be described.

We are sorry for the missing information. The legend has been modified accordingly.

2. Figure legend is too long overall. It would be better if some description in legend moves to main text.

Unfortunately, as we have added a new figure to the MS, the text is at the length limit. However, we have carefully edited the legends to remove non-essential information and to improve their readability.

3. Please, add scale bar in all figure images.

We apologise for this. Now, all figures have proper calibration bars.

4. The authors need to perform some quantification and significance analyses. In suppl Fig. 3, please quantify the frequency of GFP positive cells occurred by leakage event. In line 159, 160, authors described 'the frequency of recombinant cells with uncertain status is 'massively' reduced'. They need to show its significance. Likewise, in line 172, 'considerable number' needs to be corrected specifically. Also, in line 178, 179, quantitative analysis of mean fluorescence intensity needs to be added.

We have not quantified the frequency of GFP positive cells in Supp. Fig 3 since their presence was enough to impose a different design of the sensor.

A statistical significance has been added to the caption in Fig 1f to give statistical information to the statement of lines 159-160 (now lines **157-158**).

Line 172 (now line **174**) has been corrected accordingly.

We have addressed the point relative to lines 178-179 (now lines **181-182**) by adding the proper Kolmogorov-Smirnov tests in the in Fig. 2 caption.

5. There are some typos. In line 66, *cognitive disorders ad epilepsy -> cognitive disorders and epilepsy*. In Fig. 2d legend, maybe 'black symbols on bottom' and 'Red symbols on top'. In suppl Fig. 4 legend, there is no suppl Fig. 4g!

We have carefully revised the text.

6. In line 187-198, explanation on tamoxifen inducible Cre system is too long. Maybe authors can cut a long story short.

Agreed. We reduced this section (indeed, it was too long).

7. In Fig 3d, authors described that no Cre+ cells were found in non-amplifying group. However, as they explained well about tamoxifen-inducible Cre in line 187-198, this Cre variant is localized in cytoplasm in absence of the activator, remaining its catalytic activity. So, in Non-amplifying group in Fig. 3d, we thought electroporated neurons should exhibit cytoplasmic Cre signal at least.

As explained in the text, these experiments have been performed with a very low concentration of ER^{T2}-Cre-ER^{T2} plasmid in order to attain a very low expression of the enzyme and to reduce, in turn, the leakage effect. This low concentration may explain the detection failure. A few years back, we tested the kinetics of tamoxifen activation on a strong expressing CaMKII_Cre-ER^{T2} mouse strain. The attached figure shows that nuclear Cre is transiently visible after 24 hours from activation, but it is virtually invisible before activation and hardly detectable 36 hours after. These data are consistent with what has been shown by others in terms of timing (ref. 34 in the MS, Jahn, H. M. *et al. Sci. Rep.* 2018).

In absence of tamoxifen Cre is diluted in the entire extranuclear pyramidal cell volume, and this may be considered as a general issue in detecting tamoxifen inducible proteins.

In addition, in the specific case shown in Fig. 3d, a further possible explanations for the low Cre signature away from stimulation regards the fact that we used ER^{T2}-Cre-ER^{T2} rather than a simple Cre-ER^{T2}. Possibly, in absence of tamoxifen, the double bonding with HSP90 (at both N- and C- terminus of the enzyme), may hinder the epitope recognition by the antibody. This is mostly important in consideration of the fact that our anti-Cre is a monoclonal IgG1 (Sigma-Aldrich, C7988).

8. In line 270, rationale for why authors use 1:80 molar ratio needs to be added.

This ratio was chosen on the basis of the dose-effect relationship shown in Fig 2f. We explained this in the text lines **271-273**.

9. In line 286, explanation on Supple Fig 5 b,c is came up after Supple Fig 5d. Supple Fig. 5

The figure has been rearranged accordingly. Please, note that the Supplementary figure 5 of the original submission is now supplementary figure 6.

10. In line 290 and supple Fig. 5, adding some distinct markers pointing morphological abnormalities would be helpful.

Arrowheads underlining morphological anomalies (somatic spines and filopodia) have been now added to Supplementary Fig. 6 (formerly Supplementary Fig. 5) in panels a and c. Please note that a different arrangement of the panels has been adopted for this figure.

11. In Fig 5b, gray colored PTEN signal looks like stronger in KO neuron rather than that of WT. Colocalization analysis and quantification of immunostaining signal need to be added like Fig. 4b.

As figure 5 shows clearly, the transfected population is very sparse which required very thick slices in order to image a sufficient number of neurons integrally. Furthermore, in our hands the penetration of the antibody was limited at the surface of the slice so the number of stained cells in this preparation is rather small and this precludes an accurate quantification. Most of the dark spots visible in the picture are the optical sections of blood vessels.

12. In line 311, authors described that mutated granule cells in Fig.5 also presented somatic spines. Additional magnified images showing the somatic spines would be helpful like supple Fig. 5b.

We removed the reference to somatic spines in the text since we have observed that rare somatic spines are also present on the soma of WT granule cells. In our PTEN mosaics, the density of somatic spines is increased similarly to the density increase observed in the basal dendrites, but we think that this is not worth further investigation.

13. In line 313-320, explanation on MeCP2 gene and Supple Fig. 6 is insufficient. Also, in supple Fig. 6, it would be better if supple Fig. 6d moves in front of other figures, considering the context.

The Reviewer is right, and we added some background at lines **187-196**. Please notice that the description of these experiments has been moved before of the section relative to inducible Cre.

We reformatted the figure as requested.

14. In line 167, it was written that 'Images are maximum projections obtained from 15 sections imaged every 10um from the surface (Fig 2b).' However, considering the context, the number of sections needs to be corrected to 25, not 15.

We are grateful that the Reviewer has spotted this error. This has been fixed.

15. In line 252, they need to described the data or cite the reference; the recombinase activity was temporally confined to the 24-36h following the injection.

The data is contained in Ref. 34 that is quoted around this line (now moved at the end of line 254 for better clarity).

Reviewer #2

The authors describe a modification to the Cre-dependent reporter systems commonly used for marking specific populations of cells in vivo. Specifically, they have altered a reporter that switches from RFP to GFP expression upon Cre-dependent inversion of the reporter cassette orientation to give additional expression of the Cre gene from the reporter following this inversion. This additional expression of Cre from the reporter amplifies the Cre recombinase activity in that cell ensures complete recombination of all of the reporter alleles within that cell such that a single cell exhibits only RFP or GFP expression. Standard red/green reporters can exhibit mixtures of red and green if the Cre activity in a cell is too low to recombine all of the reporter alleles in that cell. This is valuable because it reduces ambiguity in the analysis of mosaic populations where it is desirable to assess the phenotype of two distinct populations of cells.

Specifically, the new reporter construct, called Beatrix, splits the Cre gene into two fragments that, after recombination, are brought together (separated by an intron) to allow expression. This strategy reduced background Cre expression from including an inverted full-length Cre gene in the reporter.

In this manuscript, the authors describe the generation and testing of the Beatrix reporter, and show that it is useful for phenotypic analysis of two distinct neuron populations in two different brain regions of a Pten KO mosaic mouse model. They show that this strategy results in dramatically fewer “yellow” cells with co-expression of both reporter markers, as clearly shown in panels 1e, 1f for cell culture experiments, and in panels 2d-f for in vivo neuron labelling in mouse brain. The modified reporter also works well with tamoxifen-inducible Cre systems, dramatically boosting Cre expression after a single administration of tamoxifen. I'm not sure how interesting the biology of the PTEN mosaic model they demonstrate in fig. 4 (cortex) and fig. 5 (olfactory bulb) is (because it's not my field), but they clearly show that they can carry out detailed phenotypic analysis of the two mosaic populations of neurons marked in vivo with Beatrix.

The Cre reporter engineering represents a novel approach that will likely be a generally useful tool for many neuroscience applications. The experiments are well-conceived, the data are clearly presented, and the manuscript is well-written. I recommend publication of this work in Nature Communications.

One possible concern is the use of green and red color schemes in all of the figures. My understanding is that a significant fraction of people are red/green colorblind and would have difficulty interpreting many of the figure panels with the current color scheme. Although I appreciate that the reporters have green and red fluorescence, the figures could easily be pseudocolored with a different color scheme (for example green/magenta) to make them more accessible.

We are aware of this problem but, unfortunately, some figures would become really hard to interpret. For example, cells expressing both RFP and GFP, that appear yellow now (and that are also referred to as “yellow cells” in the text), would appear nearly white and that would complicate the interpretation of the data described in some panels (e.g. panel 1e). The same applies to all Green-Yellow-Red bars used in all plots describing the hue of each cell as the polar angle θ . However, if the Journal will require these changes we will happily comply.

Reviewers' Comments:

Reviewer #1:

Remarks to the Author:

1. As the frequency (or burden) of mutation could impact the severity of the neurodevelopmental disorders caused by somatic mutations [D’Gama, Alissa M., et al. Cell reports (2017), Nguyen, Lena H., et al. Journal of Neuroscience (2019)], Beatrix is able to get the attention of this field by its one of distinct feature, Tunable mosaic generation, as well as the high sensitivity of Cre reporter. In line with previous major concern #1, though the authors calculated the fraction of cells expressing only GFP among electroporated neurons in Fig. 2f, our question of what percentage of mutation-carrying neurons in the brain was modeled by Beatrix with various molar ratio remains unanswered yet. The authors may easily answer this by calculating the number of GFP-positive neurons/the number of DAPI-positive cells in mouse brain slices.

2. Through several electrophysiological analyses in Fig. 6, authors endeavored to show the applicability of Beatrix modeling pathophysiological symptoms in neurodevelopmental disorders. As authors explained, they performed these experiments and compared PTEN mosaic mice with wild-type mice. However, Fig. 6a makes us a bit confused. Do the colors of the lines in graphs or electrophysiological traces in Fig.6 matched with the color of Fig.6a illustrate (Green : Ipsilateral cortex of PTEN mosaic mice, Black : Contralateral cortex PTEN mosaic mice)? They need to clarify what the control means in Fig. 6 (contralateral cortex of PTEN mosaic mice or ipsilateral cortex of wild-type mice). Additionally, if authors showed and compared with traces of contralateral cortex in PTEN mosaic mice only, they also need to show the traces of wild-type mice’s both side of cortex, as recent study demonstrated that brain somatic activation of MTOR signaling functionally increased the excitability of neurons in the contralateral cortex [Martina Proietti Onori, et al. 2020. bioRxiv. Doi: <https://doi.org/10.1101/2020.07.08.189399>].

3. We understand that the PTEN Beatrix model in this paper may not show spontaneous behavioral seizures, but the author’s response, “Coherently with the fact that type 2 focal cortical dysplasia in patients is not characterized by frequent seizures, but by subtle EEG signs (Guerrini et al. Epilepsia, 2015), we did not observed generalized seizures but two anomalies of great pathophysiological significance.”, seems to be inappropriate. Focal cortical dysplasia type II is an important cause of intractable focal epilepsy. Two-thirds of patients who have undergone surgical resection achieve seizure free, but one third continue to have seizures (Frauser et al, Brain. 2004 Nov;127:2406-2418.) Certainly, the most important and major phenotype of FCD type II is intractable epilepsy, not subtle EEG signs (Blumcke et al Epilepsia. 2011 Jan;52:158-174). There are many previous papers (from the Baulac group in France, the Bordey group in Yale, and the Lee group in Korea) demonstrating that mouse model of FCD type II based on somatic mutations of MTOR pathways genes showed definite spontaneous behavioral seizure.

1) As the frequency (or burden) of mutation could impact the severity of the neurodevelopmental disorders caused by somatic mutations [D’Gama, Alissa M., et al. *Cell reports* (2017), Nguyen, Lena H., et al. *Journal of Neuroscience* (2019)], Beatrix is able to get the attention of this field by its one of distinct feature, Tunable mosaic generation, as well as the high sensitivity of Cre reporter. In line with previous major concern #1, though the authors calculated the fraction of cells expressing only GFP among electroporated neurons in Fig. 2f, our question of what percentage of mutation-carrying neurons in the brain was modeled by Beatrix with various molar ratio remains unanswered yet. The authors may easily answer this by calculating the number of GFP-positive neurons/the number of DAPI-positive cells in mouse brain slices.

We understand the point the reviewer raised and we want to clarify that in our intent Figure 2 aimed to show the operation of Beatrix on the *in vivo* mosaic features as result of the molecular design, independently of technical parameters that are strictly related to the *in utero* electroporation method. Indeed, the extent of the transfected area and the number of infected neurons depends on several parameters, such as plasmid concentration, injected volumes, embryonic day, position and dimensions of electrode plates (double or triple electrode), current parameters (voltage, duration and number of pulses). Therefore, the reason we decided to show here the tunability as a function of the transfected cells is that, by counting the number of GFP positive cells over the total neurons in the cortex or mouse brain, we would have unavoidably introduced in our data a variability factor that is intrinsically dependent on the IUE parameters and yield, and not attributable to Beatrix itself.

However, we completely agree with the reviewer that from the disease modelling point of view, the total number of KO (GFP positive) cells plays a pivotal role in shaping the phenotypical features of the model itself, and we regret that we did not already include this information in the study. To address this point, we followed the Reviewer advice by performing the immunostaining of representative sections from 3 mice used for the data shown in fig. 4 and 6, with an antibody against the neuronal marker NeuN. In this way we have estimated the fraction of neurons targeted by *in utero* electroporation and of the neurons carrying the mutation in cortical layers 2/3. A cortical section showing the NeuN, GFP and RFP fluorescence is shown in the new **Supp Fig. 6**. These data are reported in the main text at lines **273-277**.

2. Through several electrophysiological analyses in Fig. 6, authors endeavored to show the applicability of Beatrix modeling pathophysiological symptoms in neurodevelopmental disorders. As authors explained, they performed these experiments and compared PTEN mosaic mice with wild-type mice. However, Fig. 6a makes us a bit confused. Do the colors of the lines in graphs or electrophysiological traces in Fig.6 matched with the color of Fig.6a illustrate (Green: Ipsilateral cortex of PTEN mosaic mice, Black: Contralateral cortex PTEN mosaic mice)? They need to clarify what the control means in Fig. 6 (contralateral cortex of PTEN mosaic mice or ipsilateral cortex of wild-type mice).

This figure contained both data comparing ipsilateral-contralateral cortices of PTEN mosaic mice (panels a, b, and e, f), as well as a separate set of double registrations on WT mice as a reference control group (panels c, d). We agree with the reviewer that the color code used in these panels could appear confusing, and the figure has now been edited for clarity as requested.

Additionally, if authors showed and compared with traces of contralateral cortex in PTEN mosaic mice only, they also need to show the traces of wild-type mice’s both side of cortex, as recent study demonstrated that brain somatic activation of MTOR signaling functionally

increased the excitability of neurons in the contralateral cortex [Martina Proietti Onori, et al. 2020. bioRxiv. Doi: <https://doi.org/10.1101/2020.07.08.189399>].

In this comment, the reviewer rises a very interesting point, relative to the possible interference of the mutated focus on the contralateral site. We were aware of this possibility, as we recently described the interference of pharmacologically induced focal interictal activity on the structure of slow wave oscillations and on visual processing ([doi:10.1038/srep40054](https://doi.org/10.1038/srep40054)). Indeed, in some of the epileptiform bursts we recorded, we have observed altered activity also in the contralateral side. This propagation is subtle, not always present, and clearly demonstrated only by the study of the cross spectra of the two channels. Given the limited space at our disposal, we wish to leave this analysis to a follow up paper. However, we observed a pretty strong reduction of slow wave oscillations in the site symmetric to the PTEN-KO focus, and we think that this data addresses the point raised by the Reviewer and fits nicely in the MS. Therefore, we have now added a new panel in figure 6 (**Fig. 6e**) that compares the power of the slow wave oscillations measured in the **control** hemisphere of the PTEN mosaic mice with the matched controls. These new data show that the presence of the mutation causes a substantial reduction of delta power compared to controls indicating a certain degree of interhemispheric interference. These data are described at lines **341-345** in the Results section. We have also added the very interesting and pertinent reference indicated by the Reviewer.

3. We understand that the PTEN Beatrix model in this paper may not show spontaneous behavioral seizures, but the author's response, "Coherently with the fact that type 2 focal cortical dysplasia in patients is not characterized by frequent seizures, but by subtle EEG signs (Guerrini et al. Epilepsia, 2015), we did not observed generalized seizures but two anomalies of great patho-physiological significance.", seems to be inappropriate. Focal cortical dysplasia type II is an important cause of intractable focal epilepsy. Two-thirds of patients who have undergone surgical resection achieve seizure free, but one third continue to have seizures (Frauser et al, Brain. 2004 Nov;127:2406-2418.) Certainly, the most important and major phenotype of FCD type II is intractable epilepsy, not subtle EEG signs (Blumcke et al Epilepsia. 2011 Jan;52:158-174). There are many previous papers (from the Baulac group in France, the Bordey group in Yale, and the Lee group in Korea) demonstrating that mouse model of FCD type II based on somatic mutations of MTOR pathways genes showed definite spontaneous behavioral seizure.

We thank the Reviewer for engaging in this conversation and we realize that we have not described our phenotype accurately. When we wrote that we have not observed generalized seizures we meant that, in our experimental conditions, the epileptiform activity we observed did not propagate from the occipital cortex (the site of the mutation focus) to the motor cortex with a consequent motor readout. However, our recordings have been performed under anesthesia and we have not performed any recordings in behaving mice, as this will be matter for a follow up study. We have now modified the text and added a few references to provide a better context for our data (lines **317-325** and in the final part of the discussion).

Reviewers' Comments:

Reviewer #1:

Remarks to the Author:

The authors satisfactorily addressed my concerns. The manuscript is strengthened by the clarifications provided and the new data incorporated in this revised version.